# Critical role of electrons in the short lifetime of blue OLEDs

Jaewook Kim [1,4], Joonghyuk Kim [2,4], Yongjun Kim[1], Youngmok Son[2], Youngsik Shin[2], Hye Jin Bae[2], Ji Whan Kim [2], Sungho Nam[2], Yongsik Jung [2], Hyeonsu Kim[1], Sungwoo Kang [1,3], Yoonsoo Jung[1], Kyunghoon Lee[1], Hyeonho Choi [2] ✉ & Woo Youn Kim [1] ✉

Designing robust blue organic light-emitting diodes is a long-standing challenge in the display industry. The highly energetic states of blue emitters cause various degradation paths, leading to collective luminance drops in a competitive manner. However, a key mechanism of the operational degradation of organic light-emitting diodes has yet to be elucidated. Here, we show that electron-induced degradation reactions play a critical role in the short lifetime of blue organic light-emitting diodes. Our control experiments demonstrate that the operational lifetime of a whole device can only be explained when excitons and electrons exist together. We examine the atomistic mechanisms of the electron-induced degradation reactions by analyzing their energetic profiles using computational methods. Mass spectrometric analysis of aged devices further confirm the key mechanisms. These results provide new insight into rational design of robust blue organic light-emitting diodes.

Phosphorescent organic light-emitting diodes (PhOLEDs) are widely used in display and lighting applications due to their low power consumption and high brightness[1–3]. Despite these advantages, their operational stability is a critical problem. In particular, the lifetime of blue PhOLEDs is approximately 10–40 times shorter than those of red and green PhOLEDs and is not yet satisfactory for commercialization[3–7]. The short operational lifetime of blue devices is mainly attributed to vulnerable emitters. The blue devices have a high triplet exciton energy (2.6–3.0 eV), which opens various degradation paths that are hardly accessible in red and green devices[8,9].

Much effort has been made to prevent the degradation of blue PhOLEDs caused by the high exciton energy. Excitons tend to be created on one side of an emissive layer (EML) due to the imbalanced polaron transport in the EML[10–12]. The resulting high exciton concentration promotes bimolecular processes such as triplet–triplet annihilation (TTA), which generate highly energetic excited states and subsequently trigger degradation reactions[9,13]. Tuning of the frontier orbital energy of host materials[14,15] or the use of hybrid host materials[11,16–19] has been applied to suppress the exciton concentration. Introducing exciton management molecules[9,20] or fluorescent dyes[21–23] into the EML is also a practical solution to control the highly energetic states. In addition, various strategies derived from atomic-level understanding have been applied to avoid exciton-induced degradation reactions[11,19,24–29]. Although these efforts substantially improved the lifespan of blue OLEDs, they still do not reach the desired level for commercialization, indicating that regulating the exciton concentration is not sufficient to eliminate all significant degradation reactions.

Polarons can be another source of the degradation of blue PhOLEDs. Several experiments have reported that triplet–polaron annihilation (TPA) can play a key role in the short lifetime[30–33]. Computational studies using the kinetic Monte Carlo method[34–36] or kinetic equations[7,30,37] also revealed that TPA may significantly contribute to the degradation. For instance, polarons can generate exciton quenchers and participate in the bleaching of emitters via a degradation reaction of consisting materials[38–42]. While a high concentration of holes can instigate radical reactions, bond strength analysis indicates

[1]Department of Chemistry, KAIST, 291 Daehak-ro, Yuseong-gu, Daejeon 34141, Republic of Korea. [2]Samsung Advanced Institute of Technology, Samsung Electronics Co., Ltd., 130 Samsung-ro, Suwon-si, Gyeonggi-do 16678, Republic of Korea. [3]Present address: Innovation Center, Samsung Electronics Co., Ltd., 1 Samsungjeonja-ro, Hwasung-si, Gyeonggi-do 18448, Republic of Korea. [4]These authors contributed equally: Jaewook Kim, Joonghyuk Kim. ✉e-mail: hono.choi@samsung.com; wooyoun@kaist.ac.kr

that most covalent bonds in OLED materials are weakened in the presence of electrons, whereas holes make them stronger except C−P bonds[38,43,44]. This suggests that electrons are the primary factor behind irreversible degradation reactions, resulting in the creation of more exciton quenchers, charge traps, and further luminance drop. However, whether TPA is indeed a key cause of the short lifetime of blue PhOLEDs under real operational conditions is still unclear because it has only been studied through model devices or computations[7]. Moreover, the atomistic mechanism of electron-induced degradation of phosphorescent emitters has never been studied[45].

Here, we show that electrons play a critical role in the degradation process of Ir-based blue PhOLEDs. Our control experiments showed that exciton-only environments could not explain the lifetime trend of blue PhOLEDs under real operational conditions. Instead, electron-induced degradation significantly affects the device's lifetime. The atomistic mechanisms of various degradation reactions caused by either excitons or electrons were examined through a quantum chemical approach. Apart from the previously known exciton-induced degradation paths, we discovered new degradation paths induced by electrons or exciton-electron interactions (Fig. 1). As direct evidence of the proposed mechanism, we showed that the masses of the primary and secondary products of the new degradation reactions coincided with peaks in the mass spectrum of an aged device. These results manifest that electron-induced degradation is key to the short lifetime of blue OLEDs under real operational conditions.

## Results

### Model study on the operational lifetime of blue OLEDs

To investigate key factors of the short lifetime of blue OLEDs under real operational conditions, we prepared three experimental setups, as depicted in Fig. 2a. The device structure and fabrication process can be found in Figs. S1 and S2 and the "Methods" section. The first setup involves a whole device (WD) in which both electrons and holes are injected into its EML. We used 3',5-di(9H-carbazol-9-yl)-[1,1'-biphenyl]-3-carbonitrile as a host in the EML[46]. This molecule is known to be an electron transport type of molecule, so the faster electron transport gives rise to an electron-dominant environment over holes in the EML[37]. Thus, we aimed to design model devices to mimic such conditions. The second setup involves an electron-only device (EOD) irradiated by an LED lamp without electron injection, which was designed to study the effect of excitons only. The third setup allows electrons to flow in addition to UV irradiation, enabling us to examine the effect of either electrons or excitons. We measured the operational lifetimes of

the two model devices and compared the results with the lifetime of the WD under real operational conditions.

We adopted facial homoleptic tris-cyclometalated iridium (III) complexes with two types of ligands for blue emitters: imidazole for Emitters 1 and 2[26] and N-heterocyclic carbene (NHC) for Emitters 3[25] and 4[15], as shown in Fig. 2b. We examined the characteristics of each device, and the results are shown in Figs. S3–S7, Tables S1 and S2. All emitters have similar peak emission wavelengths of 460–470 nm and frontier orbital energies of −5.3 eV for the highest occupied molecular orbital (HOMO) and −2.7 eV for the lowest unoccupied molecular orbital (LUMO). The J–V characteristics of all devices are similar to each other, meaning that their charge balances are also similar to each other[47,48]. Therefore, the ratios between excitons and polarons in each WD are also expected to be similar.

The lifetimes of WDs were measured in the constant current mode. Although each device has a similar peak emission wavelength, the overall emission spectra are different. For a fair comparison, their lifetimes were measured under a condition where each device emitted the same amount of energy. The initial radiance was set to 2000 mW/sr/m², ensuring that all devices had a brightness level of 300–600 cd/m².

For a direct comparison between EODs and WDs, key conditions affecting the degradation of the emitter molecules should be the same for both setups. However, making all individual conditions identical is impossible because the EODs have no hole injection, and more importantly PL and EL have different exciton density profiles inside the EML. Nevertheless, operating conditions can be controlled so that the impact of polarons and excitons on the device lifetime is similar in both setups. To this end, we imposed the following two conditions on the devices. First, we ran the same amount of current through the EODs as that of the WDs to ensure that electrons have the same effect on the device's lifetime. As described above, we minimized the effect of holes in the WDs by using the electron transport type of host molecules[46]. As a result, most electrons and excitons were at the interface of the exciton blocking layer and the EML, which has been measured using an exciton probe layer (Supplementary Discussion and Fig. S8)[10]. Second, we carefully adjusted the intensity of an LED lamp shining on the EODs, so each EOD had a similar lifetime to that of the corresponding WD at the same current. Imposing this condition was not to make the same exciton density profiles of the EODs as the WDs. Polarons (electrons in our setup) and excitons are the primary variables affecting the lifetime. Since we made the effect of electrons similar by applying the same currents to the EODs and WDs, we can

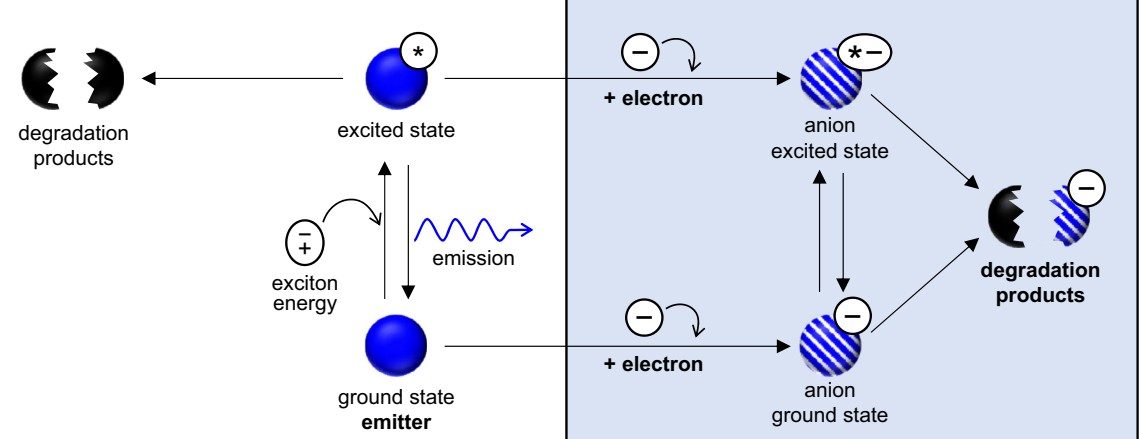

**Fig. 1 | Electron-induced degradation paths.** Various degradation reactions of excited and neutral blue emitters are caused by either excitons or electrons.

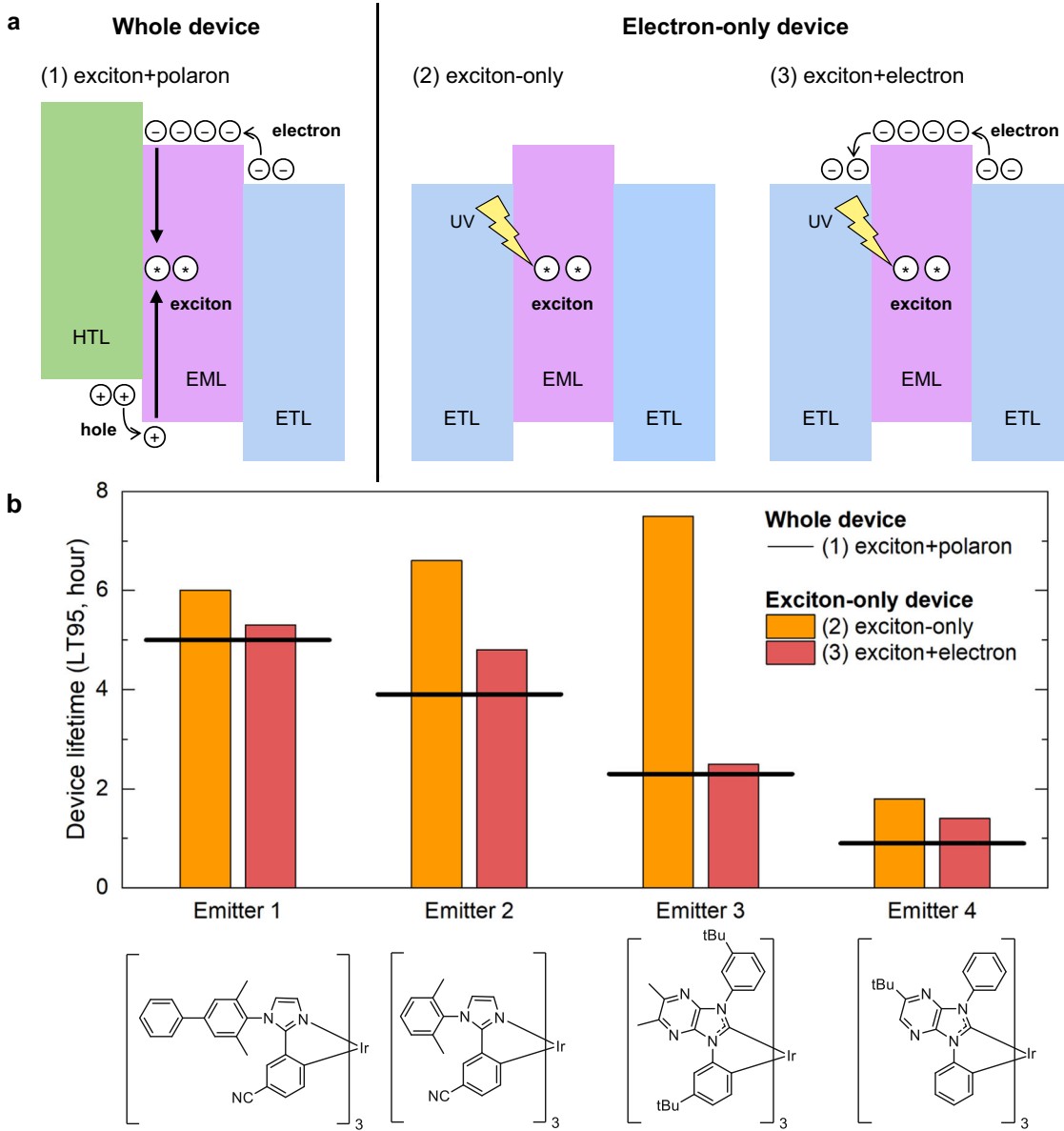

**Fig. 2 | Control experiments with model devices and their lifetimes. a** Three device setup for control experiments: (a1) a complete electroluminescent device, (a2) an electron-only device with UV illumination, and (a3) an electron-only device with both UV illumination and electric current. HTL and ETL denote hole and electron transport layers, respectively. UV indicates ultraviolet light. **b** Operational lifetimes of four blue Ir emitters measured in the three experimental setups.

expect the overall effect of excitons on the lifetime to be similar if we equalize their lifetimes. However, the exciton-only EODs have no electrons, so they would have different lifetimes under the same excitation condition as that of the exciton + electron EODs. Therefore, we can investigate how excitons with and without electrons affect the lifetime of the EODs and compare it to the WDs. The operating conditions for each device are given in Tables S3 and S4.

Figure 2b shows the lifetimes of all devices, which were measured until the luminance of each device decreased to 95% of the respective initial value (LT95). The lifetime of exciton-only EODs follows a different trend than that of exciton + electron EODs. For devices with emitters 1 and 4, the lifetimes of exciton-only EODs and exciton +electron EODs are similar. However, for devices using emitters 2 and 3, the lifetimes of the EODs differ significantly depending on the presence of electrons. Notably, for the exciton-only EOD, Emitter 3 has a longer lifetime (7.5 h) than Emitter 1 (6.0 h), whereas the WD lifetime of Emitter 3 (2.5 h) is shorter than that of Emitter 1 (5.3 h). This indicates

that the effect of excitons only is not sufficient to explain the lifetime under real operational conditions.

**Atomistic mechanism of the degradation process of Ir emitters**
Since every device uses the same structure and materials, except for the emitter, the degradation of the Ir emitter will be a key cause of the difference in operational lifetime. To elucidate the key mechanism of the degradation reactions of Ir emitters, we performed computational analysis of various possible reaction paths triggered by excitons and electrons. We examined how vulnerable all emitters are to each path by evaluating their activation barriers and bond dissociation energies (BDEs) (see Tables S5 and S6). For comparison, we also considered previously reported exciton-induced degradation reactions. For all emitters, ligand-breaking reactions induced by an exciton and ligand-releasing reactions induced by the TTA process were identified as thermodynamically accessible paths. In addition, we newly discovered three electron-induced degradation paths, as depicted in Fig. 3a.

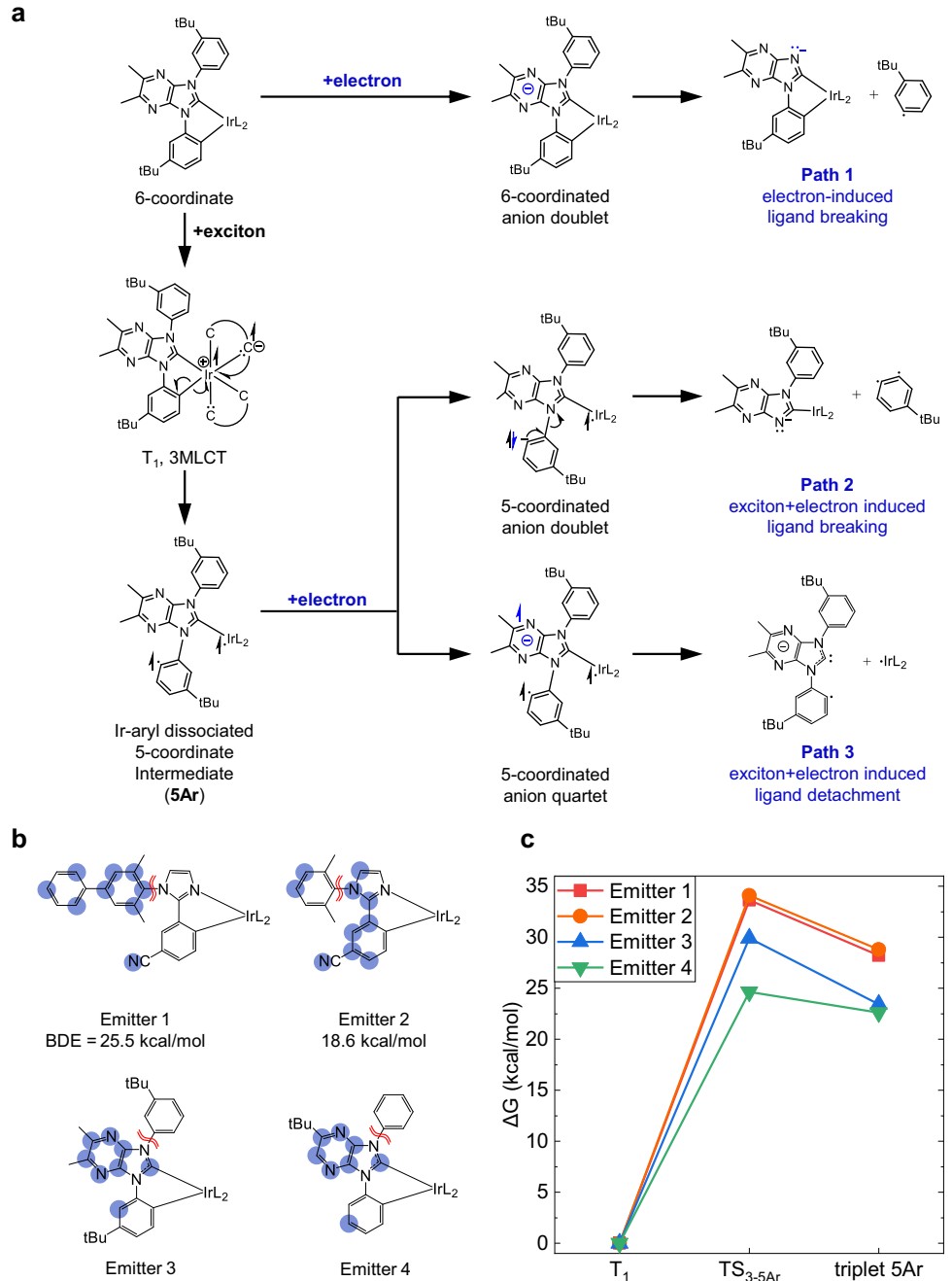

**Fig. 3 | Electron-induced degradation reaction of Ir emitters. a** Three electron-induced degradation paths of Emitter 3 and their products. **b** Distribution of one additional electron in each anion emitter molecule (blue shaded regions, $\Delta N_{electron} > 0.01$) and bond dissociation energy (BDE) of the red-marked C–N bond of each emitter. **c** Relative Gibbs free energy profile of the Ir-arylic carbon bond breaking reaction of triplet emitters.

Path 1 is an electron-induced ligand-breaking reaction. An additional electron in an emitter molecule makes it fragile by stabilizing degradation products. For instance, the C–N bond length rarely changes with the additional electron, but the C–N bond is significantly weakened when its BDE becomes less than 30 kcal/mol (Tables S5 and S7). As a result, C-N bond cleavage can occur via thermal activation.

Paths 2 and 3 require one exciton and one electron to sequentially break two chemical bonds. First, the emitter is excited to the lowest triplet state, leading to an electron transfer from a metal d-orbital to the NHC moiety ($T_1$ state, 3MLCT). The migrated electron weakens the metal-ligand bond in the *trans* position of NHC by the *trans* effect. As a result, the bond between Ir and an aryl group (Ir–$C_{Ar}$ bond) becomes vulnerable to the homolytic bond cleavage, yielding a 5-coordinated

intermediate (5Ar). Note that the 5Ar intermediate differs from the previously known 5-coordinated intermediate created by the thermal population of a triplet metal-centered state (3MC)[13,49]. The unpaired electrons of 5Ar are located on the arylic carbon and Ir, whereas the two unpaired electrons of 3MC are on Ir. Second, an electron is added to 5Ar and destabilizes it by weakening one of its bonds depending on the spin of the added electron. Path 2 is favored if the additional electron has a spin opposite to that of the electron on the arylic radical of 5Ar. It stabilizes the radical by pairing with the unpaired electron, leading to a 5-coordinated anion doublet intermediate (Fig. 3a). This triplet-electron annihilation process is energetically favored. Subsequently, the C–N bond can be readily broken by a mechanism similar to that in Path 1. Path 3 occurs if the additional electron has the same spin

as the unpaired electron on the carbon atom. In this case, three unpaired electrons coexist in the anion 5Ar intermediate. The resulting anion quartet state is highly energetic and thus likely to induce a ligand detachment reaction, as described in Fig. 3a.

The effect of each degradation path on the stability of a blue emitter varies depending on its structure. In the case of Path 1, our calculation results show that Emitter 2 (BDE = 18.6 kcal/mol) is particularly vulnerable to this path compared to the other emitters (BDE = ~25 kcal/mol). This difference is caused by the distribution of one additional electron in each anion emitter. Figure 3b shows the distribution of the additional electron of each emitter. Emitter 2 has a relatively high electron density on the nitrogen atom of the C−N bond. Emitter 1 has a similar structure to Emitter 2, but its LUMO is delocalized over a phenyl group that is absent in Emitter 2. The LUMOs of Emitters 3 and 4 are mainly distributed on the NHC moiety and have a node on the nitrogen atom of the imidazole moiety (see Fig. S9). To activate C−N bond cleavage, the additional electron should be placed on the nitrogen atom because it stabilizes the radical produced after homolytic bond cleavage (Fig. 3a)[38,39]. Consequently, Emitter 2, which has a higher additional electron population on the nitrogen atom, favors the C−N bond-breaking reaction.

Paths 2 and 3 are initiated by the formation of 5Ar after one of the Ir-$C_{Ar}$ bonds is broken. The ligand structure of each emitter directly affects the strength of the Ir-$C_{Ar}$ bonds. Our calculations show that the Ir-$C_{Ar}$ bonds of Emitters 1 and 2 with imidazole are more robust than those of Emitters 3 and 4 including NHC. In the case of the imidazole-based emitters, the Ir-$C_{Ar}$ bond-breaking reaction has a high energy barrier of 33 kcal/mol, whereas the NHC-based emitters have a barrier of less than 30 kcal/mol for the same reaction (Fig. 3c). This trend can be explained by the *trans* effect of NHC, which weakens the metal-ligand bond in the *trans* position of NHC. Therefore, Paths 2 and 3 are more favored for the NHC emitters (Emitters 3 and 4).

In summary, Emitter 2 is the most vulnerable to Path 1 among all the emitters, and Emitters 3 and 4 are less robust against Paths 2 and 3. Emitter 1 is relatively robust against the three electron-induced degradation paths. This result is consistent with its longest lifetime under real operational conditions (Fig. 2b). Additionally, it explains the smallest gap between the lifetimes of the exciton-only and exciton+electron EODs for Emitter 1. For the other emitters, however, the lifetime of the respective exciton + electron EOD is less than half that of the exciton-only EOD and close to that of the WD. These results strongly support that the three proposed electron-induced degradation reactions are directly related to the lifetime of blue OLEDs under operational conditions.

## Mass analysis of the degradation products of an aged blue PhOLED device

We analyzed the degradation products obtained from an aged blue PhOLED device using mass spectrometry to find experimental evidence of the proposed electron-induced degradation mechanisms. We prepared a WD using Emitter 3 as an example. Assigning all the degradation products is challenging because numerous reactions can occur while preparing the sample. Therefore, we focused on tracing the products possible from each degradation path in the mass spectrum of the aged device.

Figure 4a shows the energy profiles of Emitter 3 along the three electron-induced degradation reactions (Paths 1–3). We also considered previously known exciton-induced degradation reactions of Emitter 3: TTA-induced ligand detachment (Path 4) and an exciton-induced C−N bond cleavage reaction (Path 5). Paths 1, 2, and 5 break the C−N bond (green in Fig. 4a), giving rise to products P1 and P2 shown in Fig. 4b (green). Paths 3 and 4 detach a ligand from the emitter (red), resulting in products P3 and P4 shown in Fig. 4b (red). Since multiple paths can give identical products, we further studied

which path is more favored under real operational conditions for the given products.

For the ligand detachment reaction whose products are P3 and P4 in Fig. 4b, Path 3 is favored over Path 4 for the following three reasons. First, the lifetime of the exciton-only EOD is much longer than that of the WD. Thus, Path 4, which is the exciton-induced reaction, cannot explain the operational lifetime alone. Second, Path 4 requires two excitons at the same time, which is unlikely in real operational conditions considering that various methods are used to suppress a high concentration of excitons. In contrast, Path 3 requires one exciton and one electron. Third, the 5-coordinated intermediate of Path 4 (3MC in Fig. 4a) is kinetically less favorable than that of Path 3 (5Ar). The 3MC state is energetically more favored than the 5Ar state. However, it is known that the 3MC state tends to rapidly return to the original 6-coordinated structure due to the very low energy barrier of the backward reaction ($TS_{3MC}$, 1.49 kcal/mol)[49]. In contrast, the backward reaction of the 5Ar state has a relatively high energy barrier ($TS_{3-5Ar}$, 6.48 kcal/mol, see Table S6). Moreover, 5Ar is a radical intermediate formed by a cleavage of a metal-carbon bond. Therefore, the 5Ar intermediate is likely to precede the subsequent degradation reaction to neutralize the highly reactive radical.

The C-N bond cleavage whose products are P1 and P3 in Fig. 4b can occur through Paths 1, 2, and 5. Path 2 requires one exciton and one electron, whereas Path 1 requires only one exciton, and Path 5 requires only one electron. Therefore, Paths 1 and 5 will be more favored than Path 2. In the case of the WD, the LUMO of Emitter 3 has lower energy than that of the host molecule used in this work, so the emitter can readily accommodate an additional electron from the host similar to an electron trap (Fig. S1). Thus, Path 1 is more favored than Path 5.

We first investigated the primary products of the degradation reactions (P1–P4). We found the corresponding peaks in the mass spectrum measured for the EML of an aged device and tracked the changes in their intensities as the device gradually degraded. Figure 4c shows that the peak intensities monotonically increase, meaning that the amounts of all degradation products increase as the device ages. For example, the peak intensity corresponding to P3 increases by 5.7% for the LT50 device and 8.8% for the LT30 device. The isotope distributions of the iridium-containing primary products P2 and P4 were further analyzed. The result shows that both degradation products assigned as P2 and P4 ($m/z$ = 1293.63, 1015.47) contained iridium atoms, proving that P2 and P4 were the degradation products of the emitter molecule (Fig. S10). This indicates that the electron-induced degradation reactions indeed occurred inside the EML under operational conditions. These degradation reactions chemically bleach the emitter molecule, and the resulting degradation products can play as charge traps and exciton quenchers[40]. Therefore, the degradation reactions proposed here are likely to be the main cause of the irreversible degradation of the device.

We further analyzed the mass spectrum of the aged device to find the secondary products of degradation reactions. Figure 4d shows the relative intensities of the mass spectrum peaks of the LT10 device with respect to those of the LT100 device. As numerous reactions can occur inside the aged device during operation and sample preparation processes, analyzing all mass peaks is not feasible. Instead, we traced the mass peaks in the spectrum corresponding to the secondary products formed by the reaction of the primary products (P1–4 in Fig. 4b) with host molecules in the EML. The second product with a mass of 1050.52 amu can be formed by the combination of P1, P2, and the host molecule. To validate that the six identified products were the result of degradation reactions in the EML, we examined the intensity changes in the mass spectrum with respect to the depth (Fig. S11). The result shows that all the

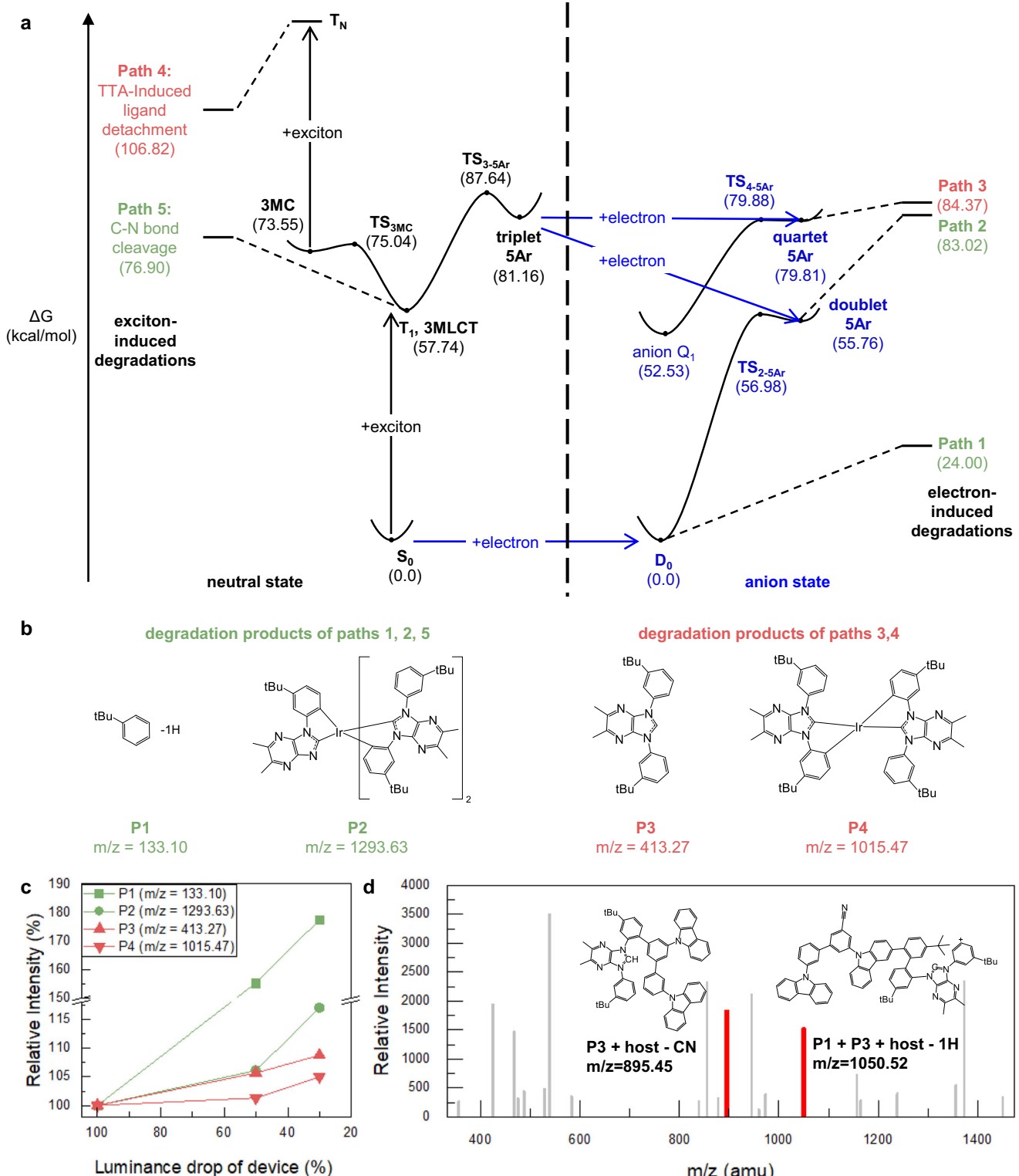

**Fig. 4 | Mass spectrometric analysis of an aged OLED device for Emitter 3.**
**a** Energy profiles for various degradation reactions of Emitter 3. **b** Primary degradation products of Emitter 3 produced along each reaction path and their mass values. **c** Relative intensities of the mass peaks of P1–P4 in (**b**). **d** Mass spectrum of the degradation products in the EML of the aged device whose luminance dropped to 10% of the initial value. Each red peak corresponds to the secondary degradation products formed by the primary products in (**b**) with a host molecule. The mass-to-charge ratio (*m/z*) is written in atomic mass units.

degradation products were found in the EML. This together with the results of the primary products provides experimental evidence for the feasibility of the proposed electron-induced degradation reactions in a blue OLED device under real operational conditions.

## Discussion

Increasing the lifetime of blue OLED devices is a difficult problem because highly energetic excitons can cause various degradation reactions. Most previous studies mainly focused on elucidating exciton-induced degradation paths. Here, we newly discovered that

electron-induced degradation reactions play a more crucial role in the degradation of blue OLED devices with Ir-based emitters under real operational conditions. Exciton- and electron-induced degradation processes may occur in a competitive manner depending on the operational conditions and component materials. Preventing one process will activate the other, which makes the development of robust blue OLEDs extremely challenging. Various practical solutions have been proposed to alleviate exciton-induced degradation, leading to substantial improvement of the lifespan of blue OLEDs, but it is far from the desired level for commercialization. Therefore, a new strategy to suppress the electron-induced degradation processes will be important for successful commercialization.

## Methods

### Device fabrication

Pre-patterned 150 nm-thick ITO-glass substrates were treated with wet cleaning (acetone, isopropyl alcohol, and deionized water) and dry cleaning (UV-ozone treatment) processes. The OLED devices (ITO (150 nm)/hole injection layer (HIL, 10 nm, p-doped (3 wt%, NDP series, purchased from Novaled AG) N-([1,10-biphenyl]−4-yl)−9,9-dimethyl-N-(4-(9-phenyl-9Hcarbazol-3-yl) phenyl)−9H-fluoren-2-amine (BCFA))/hole transporting layer (HTL, 135 nm, BCFA)/exciton blocking layer (EBL, 10 nm, oCBP)/EML (40 nm, mCBP-CN:emitter 10%)/hole blocking layer (HBL, 10 nm, mCBP-CN)/electron transport layer (ETL, 30 nm, DBFPO:LiQ = 5:5)/electron injection layer (EIL, 1 nm, LiQ)/Al (100 nm)) were fabricated by the process where the organic layers were consecutively deposited on the ITO-glass substrates at a rate of 0.1–1 Å/s and Al electrodes were thermally evaporated at a rate of 1.5 Å/s under high vacuum conditions (<1.0 × 10$^{-6}$ torr). The device area (4 mm$^2$) is defined by the overlap between the anode and cathode electrodes. The EODs (ITO/ETL/EML/ETL/MgAg (50 nm, Mg:Ag = 10:1)) were fabricated using the same materials used in the WD. All OLED devices were encapsulated in a UV-curable resin with a glass lid in an N$_2$-filled glovebox before device measurement. The films for measuring the photoluminescence quantum yield (PLQY) of emitters were fabricated using the same materials used in the EML of WDs.

### Device and film characterization

The current density–voltage–luminance (J–V–L) characteristics and electroluminescence spectra were measured with a source meter (2636B, Keithley) and spectroradiameter (SR-3AR, Topcon). PLQY of EML films was measured with a PLQY spectrometer (Quantaurus-QY, C11347, Hamamatsu Photonics). The operational lifetime of blue PhOLEDs was measured under constant current conditions at an initial radiance of 2000 mW/sr/m$^2$. The lifetime of the EODs was measured by the change in the photoluminescence intensity obtained when irradiating the device with a high-power LED lamp (LEDMOD, OMICRON) having an excitation wavelength of 340 nm. Thus, the entire area of the OLED pixel was evenly illuminated.

### Degradation product analysis

We prepared a WD using Emitter 3 and operated the device until its luminance decreased to 50%, 30%, and 10% of the initial value. Mass spectra were acquired using an OrbiSIMS (ION-TOF GmbH, Muenster, Germany) with a gas cluster ion beam (GCIB) as the primary ion beam and an orbitrap mass analyzer (Thermo Fisher Scientific, Bremen, Germany) for high mass resolution[50]. The secondary positive ions were obtained from a 200 × 200 μm$^2$ area with a DC 5 keV Ar$_{1000}^+$ ion beam (0.1 nA). The electron flood gun was needed for charge compensation. The m/z values of the compounds produced by device degradation were obtained by comparing the mass spectrum with that of the new device. The intensity of each peak corresponds to the ratio by which the number of corresponding molecules increases after the degradation of the device.

### Calculation method

Geometry optimization, enthalpy, and Gibbs free energy calculations were performed using the Gaussian 16 package[51] with the B3LYP exchange-correlation functional. We used the def2-SVP basis set for the hydrogen, carbon, and nitrogen atoms. To describe the core electron of the iridium atom, an effective core potential method with the cc-pVTZ basis set was used. The BDE was calculated from the enthalpy difference of each molecule, and the energy profiles were calculated using the Gibbs free energy differences. The thermodynamic properties were calculated at 298.15 K. The distribution of the additional electron was calculated using natural bonding orbital analysis[52]. The detailed method can be found in Supplementary Method.

## Data availability

The authors declare that all data supporting the findings of this study are available within the paper and the Supplementary Information files.

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

## Acknowledgements

This research was supported by the Samsung Advanced Institute of Technology (SAIT).

## Author contributions

Ja.K. coordinated all calculations, analyzed the data, and wrote the paper. Y.K., H.K., and S.K. performed the DFT calculations. Jo.K., J.W.K., and S.N. fabricated the devices and measured their characteristics and performance. Y. So analyzed the lifetime of electron-only devices. Y. Sh. performed the mass spectroscopic analysis of the degraded device. H.J.B. synthesized emitters. Yon.J. synthesized the host molecule. Yoo.J. and K.L. contributed to the interpretation of the degradation product analysis. W.Y.K. and H.C. conceived and organized the project. All the authors discussed the results and edited the paper.

## Competing interests

The authors declare no competing interests.
