## [Peer Review File · Nature Communications]

Critical role of electrons in the short lifetime of blue OLEDsREVIEWER COMMENTS

Reviewer #1 (Remarks to the Author):

In the present manuscript NCOMMS-22-51354-T, the authors address a very important research and development topic in the field of organic light-emitting diodes (OLEDs). They investigate degradation pathways and mechanisms in blue phosphorescent OLEDs. The limiting blue OLED lifetime is one of the most pressing questions in the field. The topic of this manuscript is therefore very well placed and of high importance. Overall, I find this an excellent manuscript, which is well placed in Nature Communications. There are some technical details I would recommend to add / revise / clarify in a revision. Please find these comments below.

Overall, the description of the experimental devices setup is not detailed enough. For this I have a couple of specific points/questions, I would like to ask the authors to work on:

1. As mentioned by the authors, the comparison made in Figure 2 only makes sense, if the same operation conditions are met on all three devices. Here, as far as I understand, the different excitation conditions are chosen such that the same luminance is achieved. However, PL and EL have slightly different excitation profiles. In EL there might be a very thin recombination zone, while the PL excitation might lead to an almost equally distributed exciton density within the EML. Depending on the EL, this may easily lead to a difference of a factor between 1 and 10 in the exciton density, the latter responsible for the exciton-electron interactions. Do you authors have a better way to calibrate or discuss the uncertainties at least?
2. The description of the PL excitation does not include the spot size of the laser illumination. Is the complete OLED pixel illuminated? Is it suffering from spatially inhomogeneous excitation because of the laser beam profile (TEM modes)?
3. What I also did not explicitly discuss/explain, which current density for the EOD with excitons and electrons are used to match the condition of the WD driving conditions. I only read that these devices were stressed at 6.25 mA/cm^2 , but how does that relate to the WD? Please add details.
4. In general, please add a detailed paragraph to explain the overall calibration procedure.

5. For completeness, I would suggest to add the respective EOD J-V characteristics for the reader. It might be interesting to check on this data.

On a more general level, one thing that might be interesting but is understandably out of scope of this current work, is the control data for hole + exciton effects. I acknowledge that the authors relate to the electron transporting host as a main reason why hole effects might be minor anyway, but this may only hold true for the specific device setup they have chosen. Maybe the authors can discuss this a bit more in introduction and/or actual manuscript discussion section.

Reviewer #2 (Remarks to the Author):

Kim et al. report the degradation mechanism of blue-phosphorescent emitting layers (EMLs) in organic light-emitting devices (OLEDs). The authors performed device experiments and computational studies for a series of homoleptic tris-cyclometalated Ir(III) complexes having imidazolato ligands or N-heterocyclocarbenic (NHC) ligands. Comparisons of the operational lifetime of devices with a charge-balanced EML or electron-only EMLs suggested the degradation mechanism that involved both an exciton and an electron. Quantum chemical calculations were employed to support this mechanism: they indicated negative polarons being the key intermediates for the intrinsic degradation of the blue-emissive Ir(III) complexes. OrbiSIMS mass analyses further provided evidence for reductively cleaved byproducts. This reviewer is supportive of this research. The advance of OLEDs has been retarded by the poor longevity of blue-emissive OLEDs. The operational stability can be improved only through understanding the intrinsic degradation mechanism of constituent materials; therefore, the research described in this manuscript should be highly acknowledged. However, this reviewer believes that the research remains fragmental and, thus, it requires substantial revisions. Technical comments are attached, as follow:

Technical comments

1. The biggest concern about this research is the lifetime (LT) data. This reviewer is reluctant to the authors' claim that the LT data of electron-only devices (EODs) can be compared with those of whole devices (WDs). Note that the former (i.e., the LT result of EODs) was based on the photoluminescence intensity decrease, whereas the latter (i.e., the LT result of WDs) was from the electroluminescence intensity decrease. The exciton generation differed from EODs and WDs. Moreover, the fluence of excitons and the flux density of electron carriers might be different. It is also emphasized that the four Ir(III) complexes exhibit the different molar absorptivity. The LT data for EODs, therefore, cannot be directly compared before suitable corrections. These disparities raise a concern about the validity of the lifetime data. Since the lifetime data guided the subsequent research in this research, they should be carefully re-examined. This reviewer recommends the authors to select alternative quantification parameters, such as the driving voltage change or the capacitance change. The light intensity is not a good parameter. Finally, this reviewer cannot find logical evidence that relates the lifetime data and the accumulation of degradation products.

2. Following the previous comment, this reviewer strongly believes that the research can be substantially improved by performing spectroscopic investigations. As commented above, the validity of this research is diminished by the illogical gap between the operation lifetime and computational proposals. The weakness can be overcome through experiments, including photolysis, electrolysis and photoelectrolysis.

3. There are numerous experimental studies which pointed to triplet-polaron annihilation as the key degradation mechanism: *J. Appl. Phys.* 2009, 105, 124514; *Org. Electron.* 2011, 12, 2056; *ACS Appl. Mater. Interfaces* 2013, 5, 8733; *J. Mater. Chem. C* 2016, 4, 8696; *Appl. Mater. Interfaces* 2017, 9, 636; *Appl. Phys. Lett.* 2017, 111, 203301; *Org. Electron* 2019, 67, 43; *Sci. Rep.* 7, 1735; *Adv. Electron. Mater.* 2019, 5, 1800708; *Sci. Adv.* 2020, 6, eabb2659. In addition, computational study by Tonnelé also suggested an occurrence of TPA (*Angew. Chem., Int. Ed.* 2017, 56, 8402). Given that, the claim that “the atomistic mechanism of polaron-induced degradation has never been studied” should be toned down.

4. To this reviewer, it is quite surprising that the excited-state 5Ar species shown Figure 3a has an unpaired electron in the phenyl ring. This structure appears to disobey the common understanding about the MLCT transition state that possesses an electron in the neutral ring of a cyclometalating ligand. It is highly recommended for the authors to show the full PES involving the initial 3MLCT transition state of the six-coordinate structure and the putative 3MLCT transition state of the pentacoordinate 5Ar species. In addition, as far as this reviewer knows, the Ir-NHC complexes usually exhibit small MLCT character in their excited states. The authors should provide deeper explanations for the generation of 5Ar.

5. The benzyne fragment in Path 2 in Figure 3a is not found in the experimentally observed mass result outlined in Figure 4b. The authors should explain this discrepancy.

6. All the structures shown in Figure 3b are incorrect. Only one of the ligands, not all three ligands, should carry spin density.

7. It is found in Figure 4a that the barrier for 5Ar to the T1 state is greater than that of the 3MC state to the T1 state. This finding led the authors to claim that degradation might be favored from 5Ar over the 3MC that relaxes rapidly. This reviewer cannot fully agree with this claim because the free energy change for the conversion from the T1 state to 5Ar is also greater than that for the conversion to the 3MC state. An equilibrium ratio among the 3MC state, the T1 state, and 5Ar should be computed to support this claim.

8. This reviewer is not fully convinced with the mass spectrometric data and their assignments shown in Figure 4d. Full analyses for the other peaks are essential. The structural assignments should also be validated by comparing isotope distributions for the experimental data and theoretical ones.

9. Finally, this reviewer carefully analyzed the manuscript, but was unable to find any conceptual difference between the proposed mechanism and the conventional mechanism involving TPA followed by degradation from hot polarons. The mechanism outlined in this manuscript is actually identical to TPA, except that reductive quenching of an exciton is proposed in place of energy transfer to a polaron. The electron-transfer quenching of excitons was also proposed previously (e.g., *Nat. Commun.* 2018, 9, 1211). The authors are, therefore, highly recommended to underscore their mechanistic novelty.

Reviewer #3 (Remarks to the Author):

This study reports the origin of the short lifetime of blue OLEDs, and the authors concluded that electrons in OLEDs are the origin of degradation. Significant efforts have been made on this issue, as the authors described in the Introduction. Studies on TTA, STA, and TPA have been carried out to clarify this problem. This study by Jaewook Kim et al. is one of these studies.

The authors say, "electron-induced degradation reactions play a critical role in the short lifetime," but they do not show experimental data on OLEDs only containing electrons. This is clear in Fig. 2; the three devices contain 1) exciton+hole+electron, 2) exciton only, and 3) exciton+electron. From here, we cannot say that the critical factor of the degradation is electrons, but rather TPA. This is not a new insight.

Moreover, if the authors' conclusion is proper, the four exciton-only devices (this does not contain electrons!) should show a very long device lifetime. However, the lifetimes are limited even for these exciton-only devices (LT95 = 2 – 10 hours even @500 cd/m², Fig. 2b). This shows that the degradation's critical factor differs from electrons.

As the authors recognize, changing only one factor without changing all the other experimental conditions is difficult. This makes the situation difficult. For example, in excitons, the singlet and triplet ratios in devices 2) and 3) are different from device 1). The spatial distribution (that is, the density of excitons) is also different. According to the preceding studies, the distinction between singlet and triplet is essential but is not discussed here.

C-P=O and C-N bonds are weak mostly for electrons (see Adv. Optical Mater. 2017, 5, 1600901 and Adv. Optical Mater. 2020, 8, 2000102). I recommend the use of other materials that are more resistant to electrons. The AOM 2017 paper also summarizes the reported device lifetimes. Even nine years ago, the lifetime was 213 h @LT80,1000 nit for an Ir-complex with similar CIE and emission wavelength (Nat. Commun. 2014, 5, 5008). It has been further updated now.

OLEDs with relatively long device lifetimes have been reported. If the authors' new finding is true, the lifetime of these devices can be further improved.

EOD means electron-only device. So, exciton-only EOD and exciton+electron EOD are meaningless.

Response to Review 1.

We thank the reviewer for the careful evaluation and valuable comments that have helped us improve the quality of our manuscript. Our point-by-point responses are attached below, and the manuscript has been revised accordingly. All changes in the manuscript were marked in blue.

In the present manuscript NCOMMS-22-51354-T, the authors address a very important research and development topic in the field of organic light-emitting diodes (OLEDs). They investigate degradation pathways and mechanisms in blue phosphorescent OLEDs. The limiting blue OLED lifetime is one of the most pressing questions in the field. The topic of this manuscript is therefore very well placed and of high importance. Overall, I find this an excellent manuscript, which is well placed in Nature Communications. There are some technical details I would recommend to add / revise / clarify in a revision. Please find these comments below.

Overall, the description of the experimental devices setup is not detailed enough. For this I have a couple of specific points/questions, I would like to ask the authors to work on:

1. As mentioned by the authors, the comparison made in Figure 2 only makes sense, if the same operation conditions are met on all three devices. Here, as far as I understand, the different excitation conditions are chosen such that the same luminance is achieved. However, PL and EL have slightly different excitation profiles. In EL there might be a very thin recombination zone, while the PL excitation might lead to an almost equally distributed exciton density within the EML. Depending on the EL, this may easily lead to a difference of a factor between 1 and 10 in the exciton density, the latter responsible for the exciton-electron interactions. Do you authors have a better way to calibrate or discuss the uncertainties at least?

Reply:

As the reviewer pointed out, EL and PL have different exciton density profiles. Therefore, the statement "the conditions applied to the EODs are very close to those of the real operating WDs" is apparently misleading. To address this concern, we redesigned the experimental setups to control the impact of both electrons and excitons in EODs as similar to that of WDs as possible, as described in the revised manuscript as follows.

1st paragraph in page 7

The lifetimes of WDs were measured in the constant current mode. Although each device has a similar peak emission wavelength, the overall emission spectra are different. For a fair comparison, their lifetimes were measured under a condition where each device emitted the same amount of energy. The initial radiance was set to 2000 mW/sr/m², ensuring that all devices had a brightness level of 300–600 cd/m².

For a direct comparison between EODs and WDs, key conditions affecting the degradation of the emitter molecules should be the same for both setups. However, making all individual conditions identical is impossible because the EODs have no hole injection, and more importantly PL and EL have different exciton density profiles inside the EML. Nevertheless, operating conditions can be controlled so that the impact of polarons and excitons on the device lifetime is similar in both setups. To this end, we imposed the following two conditions on the devices. First, we ran the same amount of current through the EODs with that of the

WDs to ensure that electrons have the same effect on the device lifetime. As described above, we minimized the effect of holes in the WDs by using the electron transport type of host molecules. [44] As a result, most electrons and excitons were at the interface of the exciton blocking layer and the EML, that has been measured using an exciton probe layer (Supplementary Section 3). [10] Second, we carefully adjusted the intensity of an LED lamp shining on the EODs, so each EOD had a similar lifetime to that of the corresponding WD at the same current. Imposing this condition was not to make the same exciton density profiles of the EODs as the WDs. Polarons (electrons in our setup) and excitons are the primary variables affecting the lifetime. Since we made the effect of electrons similar by applying the same currents to the EODs and WDs, we can expect the overall effect of excitons on the lifetime to be similar if we equalize their lifetimes. However, the exciton-only EODs have no electrons, so they would have different lifetimes under the same excitation condition with that of the exciton+electron EODs. Therefore, we can investigate how excitons with and without electrons affect the lifetime of the EODs and compare it to the WDs. The operating conditions for each device are given in Supplementary Section 4.

Fig. 2b shows the lifetimes of all devices, which were measured until the luminance of each device decreased to 95% of the respective initial value (LT95). The lifetime of exciton-only EODs follows a different trend than that of exciton+electron EODs. For devices with emitters 1 and 4, the lifetimes of exciton-only EODs and exciton+electron EODs are similar. However, for devices using emitters 2 and 3, the lifetimes of the EODs differ significantly depending on the presence of electrons. Notably, for the exciton-only EOD, Emitter 3 has a longer lifetime (7.5 h) than Emitter 1 (6.0 h), whereas the WD lifetime of Emitter 3 (2.5 h) is shorter than that of Emitter 1 (5.3 h). This indicates that the effect of excitons only is not sufficient to explain the lifetime under real operational conditions.

Figure 2b

2nd paragraph in Page 3

(...) Our control experiments showed that exciton-only environments could not explain the lifetime trend of blue PhOLEDs under real operational conditions. Instead, electron-induced degradation significantly affects the device's lifetime. (...)

2nd paragraph in page 19

(...) The operational lifetime of blue PhOLEDs was measured under constant current conditions at an initial radiance of 2000 mW/sr/m². (...)

Supplementary Information

Supplementary Section 3. Charge and exciton distribution of whole device

We used an electron transport type host molecule to minimize the influence of holes on the lifetime of the whole device (WD). [S1] In the EML of WDs, emitter molecules are doped in the host film. As the HOMO energy of the emitter molecules is higher than that of host molecules, the doped emitter molecules can improve the hole mobility in the EML. This requires a deeper investigation into the polaron distribution to ensure that the effect of the hole is well suppressed in the devices. Hence, we measured the exciton distribution in the EML by using the methodology described in the reference [S2]. We fabricated probe devices with the same structure as the WDs except an inserted thin red dye layer inside the EML. Here, Ir(dmpq)₂(acac)[S3] was co-doped at 2 vol% at different positions separated by 10 nm in the EMLs, with a doping layer width of 1 nm. The thin layer emits red light by accepting exciton energy from adjacent excited molecules. Comparing the relative intensity of red light from the probe devices, we measured the exciton densities as a function of the position of the probe layer. The relative intensity is calculated using the following equation.

$$\text{Relative Intensity}(x) = \frac{(\text{Intensity of prove device @ 612 nm})(x) - (\text{Intensity of WD @ 612nm})}{\max\{(\text{Intensity of prove device @ 612 nm})(x)\}}$$

As the charge balances of WDs are similar to each other, we fabricated probe devices for Emitter 1 and 3 as examples. Our data shows that excitons are highly concentrated near the EBL/EML interface and are almost absent in the region farther than 20 nm away from the interface. (Figure S8) The result implies that charge recombination occurs predominantly at the EBL/EML interface, and the holes will be concentrated at the same interface.

Supplementary Figure S8. Relative electroluminescence intensities of the red dopants of probe devices.

Supplementary Section 4. Operational condition of each device

Supplementary Table S3. Operational condition of WD

	Emitter 1	Emitter 2	Emitter 3	Emitter 4
Radiance ($\text{mW sr}^{-1} \text{m}^{-2}$)	2000	2000	2000	2000
Luminance (cd m^{-2})	573	561	487	350
Current (mA)	0.064	0.072	0.079	0.137

Supplementary Table S4. Operational condition of EOD

	Emitter 1	Emitter 2	Emitter 3	Emitter 4
UV lamp intensity (mW)	23.2	22.3	15	18.2
Initial radiance (cd m^{-2})	19138	18087	8991	9000
Current for exciton+electron EOD (mA)	0.064	0.072	0.079	0.137

2. The description of the PL excitation does not include the spot size of the laser illumination. Is the complete OLED pixel illuminated? Is it suffering from spatially inhomogeneous excitation because of the laser beam profile (TEM modes)?

Reply:

We appreciate for pointing out the potential problems associated with laser illumination. To address the reviewer's concern, we have used a 340 nm LED lamp in the new experimental setups. Thus, the light was uniformly illuminated over the entire pixel area.

1st paragraph in page 6

(...) The second setup involves an electron-only device (EOD) irradiated by an LED lamp without electron injection, which was designed to study the effect of excitons only. (...)

2nd paragraph in page 19

(...) The lifetime of the EODs was measured by the change in the photoluminescence intensity obtained when irradiating the device with a high-power LED lamp (LEDMOD, OMICRON) having an excitation wavelength of 340 nm. Thus, the entire area of the OLED pixel was evenly illuminated. (...)

3. What I also did not explicitly discuss/explain, which current density for the EOD with excitons and electrons are used to match the condition of the WD driving conditions. I only read that these devices were stressed at 6.25 mA/cm², but how does that relate to the WD? Please add details.

Reply:

As described in the response to comment 1, we redesigned the experimental setups and controlled the operating conditions of each device. The operating conditions for each EOD were described in response 1.

4. In general, please add a detailed paragraph to explain the overall calibration procedure.

Reply:

As described in the response to comment 1, we redesigned the experimental setups and controlled the operating conditions of each device. The operating conditions for each EOD were described in response 1.

5. For completeness, I would suggest to add the respective EOD J-V characteristics for the reader. It might be interesting to check on this data.

Reply:

We measured the J-V characteristics of the EOD and added them to the Supplementary Information.

Supplementary Information

Supplementary Figure S7. The current density-voltage curves of electron-only devices.

On a more general level, one thing that might be interesting but is understandably out of scope of this current work, is the control data for hole + exciton effects. I acknowledge that the authors relate to the electron transporting host as a main reason why hole effects might be minor anyway, but this may only hold true for the specific device setup they have chosen. Maybe the authors can discuss this a bit more in introduction and/or actual manuscript discussion section.

Reply:

It is known that the accumulation of holes in a device can act as an exciton quencher, leading to a decrease in luminance. In addition, a radical-induced reaction can occur when the concentration of holes inside the HTL is high. [*Adv. Mater.* 2023, 25(15), 2114-2129; You, Y. (2021). Chemical Mechanisms of Intrinsic Degradation of Emitting Layers in Organic Light-Emitting Devices. In: Kang, I.B., Han, C.W., Jeong, J.K. (eds) *Advanced Display Technology*. Series in Display Science and Technology. Springer, Singapore.] However, computational studies using quantum chemical methods indicate that unlike electrons, which typically weaken covalent bonds, holes actually increase the strength of most chemical bonds, with the exception of C-P bonds. [*J. Phys. Chem. C* 2014, 118(14) 7569–7578; *Chem. Mater.* 2016, 28(16), 5791–5798] Therefore, electrons are more likely to contribute to the irreversible decrease in luminance induced by the degradation of the constituent materials of EML. We have included a discussion about this point in the manuscript.

1st paragraph in page 3

(...) For instance, polarons can generate exciton quenchers and participate in the bleaching of emitters via a degradation reaction of consisting materials. [38-41] While a high concentration of holes can instigate radical reactions, bond strength analysis indicates that most covalent bonds in OLED materials are weakened in the presence of electrons, whereas holes make them stronger except C-P bonds. [38, 42] This suggests that electrons are the primary factor behind irreversible degradation reactions, resulting in the creation of more exciton quenchers, charge traps, and further luminance drop. (...)

Response to Review 2.

We thank the reviewer for the careful evaluation and valuable comments that have helped us improve the quality of our manuscript. Our point-by-point responses are attached below, and the manuscript has been revised accordingly. All changes in the manuscript were marked in blue.

Kim et al. report the degradation mechanism of blue-phosphorescent emitting layers (EMLs) in organic light-emitting devices (OLEDs). The authors performed device experiments and computational studies for a series of homoleptic tris-cyclometalated Ir(III) complexes having imidazolato ligands or N-heterocyclocarbenic (NHC) ligands. Comparisons of the operational lifetime of devices with a charge-balanced EML or electron-only EMLs suggested the degradation mechanism that involved both an exciton and an electron. Quantum chemical calculations were employed to support this mechanism: they indicated negative polarons being the key intermediates for the intrinsic degradation of the blue-emissive Ir(III) complexes. OrbiSIMS mass analyses further provided evidence for reductively cleaved byproducts. This reviewer is supportive of this research. The advance of OLEDs has been retarded by the poor longevity of blue-emissive OLEDs. The operational stability can be improved only through understanding the intrinsic degradation mechanism of constituent materials; therefore, the research described in this manuscript should be highly acknowledged. However, this reviewer believes that the research remains fragmental and, thus, it requires substantial revisions. Technical comments are attached, as follow:

Technical comments

1. The biggest concern about this research is the lifetime (LT) data. This reviewer is reluctant to the authors' claim that the LT data of electron-only devices (EODs) can be compared with those of whole devices (WDs). Note that the former (i.e., the LT result of EODs) was based on the photoluminescence intensity decrease, whereas the latter (i.e., the LT result of WDs) was from the electroluminescence intensity decrease. The exciton generation differed from EODs and WDs. Moreover, the fluence of excitons and the flux density of electron carriers might be different. It is also emphasized that the four Ir(III) complexes exhibit the different molar absorptivity. The LT data for EODs, therefore, cannot be directly compared before suitable corrections. These disparities raise a concern about the validity of the lifetime data. Since the lifetime data guided the subsequent research in this research, they should be carefully re-examined. This reviewer recommends the authors to select alternative quantification parameters, such as the driving voltage change or the capacitance change. The light intensity is not a good parameter.

Reply:

This comment is also related to the first comment from Reviewer 1. The mechanisms of exciton generation in EODs and WDs are different, so the phrase "the conditions applied to the EODs are very close to those of real operating WDs" is apparently misleading. Furthermore, Reviewer 2 pointed out that four emitters exhibit different molar absorptivity. Therefore, irradiating each EOD with the UV lamp with the same intensity will not generate the same number of excitons in each EML.

In addition, as reviewer 2 suggested, the driving voltage and capacitance changes can also be indicators of device degradation. These indicators reflect changes in charge density inside the device and provide evidence of charge accumulation.[*Org. Electron.* 2013, 14, 2518–2522] However, other factors, such

as chemical bleaching or exciton quenching, can also decrease device luminance. Therefore, we still aimed to measure the lifetime of OLED devices by examining changes in light intensity. While each device has a similar peak emission wavelength, their overall emission spectra differ. For a fair comparison, their lifetimes were measured under a condition where each device emitted the same amount of energy, i.e., the WDs have the same initial radiance. To address the concerns raised by the reviewer, we redesigned the experiment to control the impact of both electrons and excitons in EODs as similar to that of WDs as possible. In the revised manuscript, we described more details of the new device setups as follows.

1st paragraph in page 7

The lifetimes of WDs were measured in the constant current mode. Although each device has a similar peak emission wavelength, the overall emission spectra are different. For a fair comparison, their lifetimes were measured under a condition where each device emitted the same amount of energy. The initial radiance was set to 2000 mW/sr/m², ensuring that all devices had a brightness level of 300–600 cd/m². For a direct comparison between EODs and WDs, key conditions affecting the degradation of the emitter molecules should be the same for both setups. However, making all individual conditions identical is impossible because the EODs have no hole injection, and more importantly PL and EL have different exciton density profiles inside the EML. Nevertheless, operating conditions can be controlled so that the impact of polarons and excitons on the device lifetime is similar in both setups. To this end, we imposed the following two conditions on the devices. First, we ran the same amount of current through the EODs with that of the WDs to ensure that electrons have the same effect on the device lifetime. As described above, we minimized the effect of holes in the WDs by using the electron transport type of host molecules. [44] As a result, most electrons and excitons were at the interface of the electron blocking layer and the EML, that has been measured using an exciton probe layer (Supplementary Section 3). [10] Second, we carefully adjusted the intensity of an LED lamp shining on the EODs, so each EOD had a similar lifetime to that of the corresponding WD at the same current. Imposing this condition was not to make the same exciton density profiles of the EODs as the WDs. Polarons (electrons in our setup) and excitons are the primary variables affecting the lifetime. Since we made the effect of electrons similar by applying the same currents to the EODs and WDs, we can expect the overall effect of excitons on the lifetime to be similar if we equalize their lifetimes. However, the exciton-only EODs have no electrons, so they would have different lifetimes under the same excitation condition with that of the exciton+electron EODs. Therefore, we can investigate how excitons with and without electrons affect the lifetime of the EODs and compare it to the WDs. The operating conditions for each device are given in Supplementary Section 4. Fig. 2b shows the lifetimes of all devices, which were measured until the luminance of each device decreased to 95% of the respective initial value (LT95). The lifetime of exciton-only EODs follows a different trend than that of exciton+electron EODs. For devices with emitters 1 and 4, the lifetimes of exciton-only EODs and exciton+electron EODs are similar. However, for devices using emitters 2 and 3, the lifetimes of the EODs differ significantly depending on the presence of electrons. Notably, for the exciton-only EOD, Emitter 3 has a longer lifetime (7.5 h) than Emitter 1 (6.0 h), whereas the WD lifetime of Emitter 3 (2.5 h) is shorter than that of Emitter 1 (5.3 h). This indicates that the effect of excitons only is not sufficient to explain the lifetime under real operational conditions.

Figure 2b

2nd paragraph in Page 3

(...) Our control experiments showed that exciton-only environments could not explain the lifetime trend of blue PhOLEDs under real operational conditions. Instead, electron-induced degradation significantly affects the device's lifetime. (...)

2nd paragraph in page 19

(...) The operational lifetime of blue PhOLEDs was measured under constant current conditions at an initial radiance of 2000 mW/sr/m². (...)

Supplementary Information

Supplementary Section 3. Charge and exciton distribution of whole device

We used an electron transport type host molecule to minimize the influence of holes on the lifetime of the whole device (WD). [S1] In the EML of WDs, emitter molecules are doped in the host film. As the HOMO energy of the emitter molecules is higher than that of host molecules, the doped emitter molecules can improve the hole mobility in the EML. This requires a deeper investigation into the polaron distribution to ensure that the effect of the hole is well suppressed in the devices. Hence, we measured the exciton distribution in the EML by using the methodology described in the reference [S2]. We fabricated probe devices with the same structure as the WDs except an inserted thin red dye layer inside the EML. Here, Ir(dmpq)2(acac)[S3] was co-doped at 2 vol% at different positions separated by 10 nm in the EMLs, with a doping layer width of 1 nm. The thin layer emits red light by accepting exciton energy from adjacent excited molecules. Comparing the relative intensity of red light from the probe devices, we measured the exciton densities as a function of the position of the probe

layer. The relative intensity is calculated using the following equation.

$$\text{Relative Intensity}(x) = \frac{(\text{Intensity of probe device @ 612 nm})(x) - (\text{Intensity of WD @ 612nm})}{\max\{(\text{Intensity of probe device @ 612 nm})(x)\}}$$

As the charge balances of WDs are similar to each other, we fabricated probe devices for Emitter 1 and 3 as examples. Our data shows that excitons are highly concentrated near the EBL/EML interface and are almost absent in the region farther than 20 nm away from the interface. (Figure S8) The result implies that charge recombination occurs predominantly at the EBL/EML interface, and the holes will be concentrated at the same interface.

Supplementary Figure S8. Relative electroluminescence intensities of the red dopants of probe devices.

Supplementary Section 4. Operational condition of each device

Supplementary Table S3. Operational condition of WD

	Emitter 1	Emitter 2	Emitter 3	Emitter 4
Radiance (mW sr ⁻¹ m ⁻²)	2000	2000	2000	2000
Luminance (cd m ⁻²)	573	561	487	350
Current (mA)	0.064	0.072	0.079	0.137

Supplementary Table S4. Operational condition of EOD

	Emitter 1	Emitter 2	Emitter 3	Emitter 4
UV lamp intensity (mW)	23.2	22.3	15	18.2
Initial radiance (cd m ⁻²)	19138	18087	8991	9000
Current for exciton+electron EOD (mA)	0.064	0.072	0.079	0.137

Finally, this reviewer cannot find logical evidence that relates the lifetime data and the accumulation of degradation products.

Reply:

We agree that our explanation of the relationship between the degradation reaction and device lifetime was insufficient. The decrease in luminance of OLED devices can be attributed to three main factors: changes in charge balance, exciton quenching, and bleaching of emitter molecules [ACS Photonics, 2022, 9, 82-89]. Among these, the effect of charge balance on device lifetime is relatively small compared to the other two factors. The degradation reactions of the consisting materials of the EML are the main cause of the irreversible decrease in luminance of OLED devices for two reasons: the chemical reactions irreversibly deform the emitter molecule and generate exciton quenchers inside the EML. Since each device employed the same host material, the degradation of the host molecules would not be a significant factor for the discrepancy in device lifespan. Additionally, the host material is reported to be robust against polarons and excitons in the film state. [Adv. Sci., 2017, 4, 1600502] We examined change in the degradation products of the emitter molecule with respect to the operation time, which has been traced with the isotope distribution of the iridium-containing primary products. As shown in Figure 4c, we found a correlation between the increase in the amount of degradation products and the decrease in the luminance of the device.

We have revised the manuscript and included these discussions to clarify the logical evidence.

1st paragraph in page 3

(...) For instance, polarons can generate exciton quenchers and participate in the bleaching of emitters via a degradation reaction of consisting materials. [38-41] While a high concentration of holes can instigate radical reactions, bond strength analysis indicates that most covalent bonds in OLED materials are weakened in the presence of electrons, whereas holes make them stronger except C-P bonds. [38, 42] This suggests that electrons are the primary factor behind irreversible degradation reactions, resulting in the creation of more exciton quenchers, charge traps, and further luminance drop. (...)

1st paragraph in page 10

Since every device uses the same structure and materials, except for the emitter, the degradation of the Ir emitter will be a key cause of the difference in the operational lifetime. To elucidate the key mechanism of the degradation reactions of Ir emitters, we performed computational analysis of various possible reaction paths triggered by excitons and electrons. (...)

3rd paragraph in page 16

(...) For example, the peak intensity corresponding to P3 increases by 5.7% for the LT50 device and 8.8% for the LT30 device. The isotope distributions of the iridium-containing primary products P2 and P4 were further analyzed. The result shows that both degradation products assigned as P2 and P4 ($m/z=1293.63, 1015.47$) contained iridium atoms, proving that P2 and P4 were the degradation products of the emitter molecule (Fig. S10). This indicates that the electron-induced degradation reactions indeed occurred inside the EML under operational conditions. These degradation reactions chemically bleach the emitter molecule, and the resulting degradation products can play as charge traps and exciton quenchers. [40] Therefore, the degradation reactions proposed here are likely to be the main cause of the irreversible degradation of the device.

2. Following the previous comment, this reviewer strongly believes that the research can be substantially improved by performing spectroscopic investigations. As commented above, the validity of this research is diminished by the illogical gap between the operation lifetime and computational proposals. The weakness can be overcome through experiments, including photolysis, electrolysis and photoelectrolysis.

Reply:

We appreciate the valuable suggestions. The suggested experiments may reveal the mechanism of individual degradation processes. However, each experiment has different conditions from real devices, and thus the mechanism deduced from each suggested experiment may be different from what happened in the real device. After all, we showed in the revised manuscript that the predicted products from the calculations were found in the mass analysis of the real devices with the isotope analysis as explained in the previous response.

3. There are numerous experimental studies which pointed to triplet-polaron annihilation as the key degradation mechanism: J. Appl. Phys. 2009, 105, 124514; Org. Electron. 2011, 12, 2056; ACS Appl. Mater. Interfaces 2013, 5, 8733; J. Mater. Chem. C 2016, 4, 8696; Appl. Mater. Interfaces 2017, 9, 636; Appl. Phys. Lett. 2017, 111, 203301; Org. Electron 2019, 67, 43; Sci. Rep. 7, 1735; Adv. Electron. Mater. 2019, 5, 1800708; Sci. Adv. 2020, 6, eabb2659. In addition, computational study by Tonnelê also suggested an occurrence of TPA (Angew. Chem., Int. Ed. 2017, 56, 8402). Given that, the claim that “the atomistic mechanism of polaron-induced degradation has never been studied” should be toned down.

Reply:

We note that most of the references provided by reviewer 2 focus on device degradation due to hole-exciton interactions. While Giebink and his coworker discussed the degradation of phosphorescent devices by electrons, the atomistic mechanism associated with device degradation has never been provided. [*J. appl. Phys.* 2009, 105(12), 124514] Our current work aims to elucidate the atom-level mechanism of the device degradation and thus provide valuable insights to design more robust emitters. The atomistic mechanism we proposed concerns the degradation pathway described by change in the atom level of molecular structures. For example, in our 2021 paper (*Adv. Optical Mater.* 2021, 9, 2100630), we identified the dissociation of the benzylic C-H bond as the atomistic mechanism behind the degradation of an excited Ir-based blue dopant and improved the device lifetime by resolving the

problem with a new molecular structure.

To mitigate Reviewer 2's concern, our research mainly emphasizes the impact of electrons on device lifetime. Therefore, we have revised the phrase "polaron-induced degradation" as "electron-induced degradation".

1st paragraph in page 3

(...) Moreover, the atomistic mechanism of electron-induced degradation of phosphorescent emitters has never been studied. [42]

4. To this reviewer, it is quite surprising that the excited-state 5Ar species shown Figure 3a has an unpaired electron in the phenyl ring. This structure appears to disobey the common understanding about the MLCT transition state that possesses an electron in the neutral ring of a cyclometalating ligand. It is highly recommend for the authors to show the full PES involving the initial 3MLCT transition state of the six-coordinate structure and the putative 3MLCT transition state of the pentacoordinate 5Ar species. In addition, as far as this reviewer knows, the Ir-NHC complexes usually exhibit small MLCT character in their excited states. The authors should provide deeper explanations for the generation of 5Ar.

Reply:

We agree that more details are required to avoid any confusion. The 5Ar structure is generated after the triplet MLCT state as the reviewer pointed out. We have revised the manuscript to clarify the generation mechanism of the 5Ar structure accordingly.

Figure 3a

3rd paragraph in page 10

(...) First, the emitter is excited to the lowest triplet state, leading to an electron transfer from a metal d-orbital to the NHC moiety (T_1 state, 3MLCT). The migrated electron weakens the metal-ligand bond in the *trans* position of NHC by the *trans* effect. As a result, the bond between Ir and an aryl group (Ir-C_{Ar} bond) becomes vulnerable to the homolytic bond cleavage, yielding a 5-coordinated intermediate (5Ar). (...)

5. The benzyne fragment in Path 2 in Figure 3a is not found in the experimentally observed mass result outlined in Figure 4b. The authors should explain this discrepancy.

Reply:

The benzyne compound produced by Path 2 is inherently unstable and will quickly transform into P1 in Figure 4b by taking two hydrogen atoms from neighboring molecules. However, this was not clearly shown in Figures 3a and 4b. To address the reviewer's concern, we have revised these figures.

In addition, we re-examined our mass spectrometry data and identified a peak corresponding to P1. As the OLED luminance decreased, the intensity of the peak corresponding to P1 increased, further supporting our analysis.

Figure 3a

a

Figure 4b,c,d

2nd paragraph in page 16

We first investigated the primary products of the degradation reactions (P1–P4). We found the corresponding peaks in the mass spectrum measured for the EML of an aged device and tracked the changes in their intensities as the device gradually degraded. Fig. 4c shows that the peak intensities monotonically increase, meaning that the amounts of all degradation products increase as the device ages. (...)

6. All the structures shown in Figure 3b are incorrect. Only one of the ligands, not all three ligands, should carry spin density.

Reply:

The added electron occupies the emitter molecule's lowest unoccupied molecular orbital, which is uniformly distributed across the three ligands, as shown in Figure S9. Consequently, the emitter molecule retains symmetry even when it becomes an anion. (Table S7). For iridium-based phosphorescent emitters, the energies of the first three lowest unoccupied molecular orbitals are almost identical and localized on each ligand. Thus, a small perturbation will break the symmetry by pushing the electron density toward one of the three ligands. This electronic structure of the iridium-based phosphorescent emitters has been verified by spectroscopic experiments and quantum chemical calculations. [*Inorg. Chem.* 2010, 49(20), 9290-9299; *J. Comp. Chem.* 2010, 31(3), 628-638] To explain the connection between the degradation mechanism and the electron distribution, it is clearer to show a state where the additional electron is localized on one ligand, as suggested by Reviewer 2. We revised Figure 3b and added a method to determine the atomic charge in the Supplementary Information.

Figure 3b

b

2nd paragraph in page 11

(...) Fig. 3b shows the distribution of the additional electron of each emitter. (...)

3rd paragraph in page 19

(...) The distribution of the additional electron was calculated using natural bonding orbital analysis. [50] The detailed method can be found in Supplementary Section 5.

Supplementary Section

The distribution of the additional electron on the emitter molecules

The electron density distribution in the emitter molecules is determined by calculating the difference between the atomic charges in their neutral and anionic states. The electron density of each emitter molecule was obtained from its ground state structure using Gaussian16 [S4] software, and the atomic charge of each atom was calculated using Natural Bond Orbital (NBO) analysis. [S5]

The added electrons occupy the lowest unoccupied molecular orbital (LUMO) of the emitter, which is uniformly distributed across the three ligands (Fig. S9). For iridium-based phosphorescent emitters, the energies of the first three lowest unoccupied molecular orbitals are almost identical and located on each ligand. Thus, a small perturbation can break the symmetry by pushing the electron density toward one of the three ligands. This electronic structure of the iridium-based phosphorescent emitters has been verified by spectroscopic experiments and quantum chemical calculations. [S6, S7] Therefore, the atomic charge changes obtained from each ligand were summed to identify the atoms where the added electrons were located. We used a threshold of 0.01.

7. It is found in Figure 4a that the barrier for 5Ar to the T1 state is greater than that of the 3MC state to the T1 state. This finding led the authors to claim that degradation might be favored from 5Ar over the 3MC that relaxes rapidly. This reviewer cannot fully agree with this claim because the free energy change for the conversion from the T1 state to 5Ar is also greater than that for the conversion to the 3MC state. An equilibrium ratio among the 3MC state, the T1 state, and 5Ar should be computed to support this claim.

Reply:

We agree with the reviewer about that our previous claim, the lifetime of the 5Ar intermediate state is longer than the 3MC state based solely on the free energy change, was incorrect. The 5Ar is a radical intermediate that is highly reactive, so various reactions occur rapidly to resolve the radical state. Therefore, it is inappropriate to calculate the equilibrium ratio of the 5Ar state in the same way as the 3MC state, which is considered to be in thermal equilibrium with the T₁(MLCT) state. [*J. Am. Chem. Soc.* 2009, 131, 9813–9822] The 3MC intermediate is thermodynamically easy to generate but has a low activation energy for the reverse reaction. Such a low activation energy makes the 3MC intermediate kinetically unfavorable and promotes the reverse reaction to the T₁ state. It is also known that the iridium-based emitter molecules in the 3MC molecule can easily undergo non-radiative decay and return to the ground state. [*Inorg. Chem.* 2018, 57(15) 8881-8889] These non-destructive reactions annihilate the exciton energy without causing photochemical bleaching. We have revised the manuscript to include this discussion.

3rd paragraph in page 15

(...) Third, the 5-coordinated intermediate of Path 4 (3MC in Fig. 4a) is kinetically less favorable than that of Path 3 (5Ar). The 3MC state is energetically more favored than the 5Ar state. However, it is known that the 3MC state tends to rapidly return to the original 6-coordinated structure due to the very low energy barrier of the backward reaction (TS_{3MC}, 1.49 kcal/mol). [47] In contrast, the backward reaction of the 5Ar state has a relatively high energy barrier (TS_{3-5Ar}, 6.48 kcal/mol, see Table S6). Moreover, 5Ar is a radical intermediate formed by a cleavage of a metal-carbon bond. Therefore, the 5Ar intermediate is likely to precede the subsequent degradation reaction to neutralize the highly reactive radical.

8. This reviewer is not fully convinced with the mass spectrometric data and their assignments shown in Figure 4d. Full analyses for the other peaks are essential. The structural assignments should also be validated by comparing isotope distributions for the experimental data and theoretical ones.

Reply:

While acknowledging the valuable suggestion, we concern that a variety of degradation products will accumulate over time in a device undergoing extensive degradation. To analyze the degradation products in an aged device operating under realistic conditions, we captured secondary positive ions produced by sputtering an Ar_{1000}^+ ion beam on the aged device and examined the relative distribution of their m/z values. Given this approach, it is not expected that all the peaks we present were generated solely by the degradation process. Instead of assigning all peaks, we focused to find the peaks corresponding to the primary and secondary products derived from the degradation reactions of the EML materials predicted by computational chemistry, because they can be regarded as direct evidence of the proposed degradation reactions. We examined the mass peak intensities associated with the primary products in devices at different degradation levels. Our results showed a consistent increase in the amount of all primary products proportional to the degree of device degradation (Figure 4c). In addition, we detected the mass peaks corresponding to the secondary products and presented these results in Figure 4d.

We performed two additional analyses to confirm that the identified degradation products were indeed derived from the degradation reaction in the EML. First, we analyzed the depth profile of the mass spectrum and validated the presence of both primary and secondary products in the EML of the degraded device. Second, we examined the isotope distribution as suggested by Reviewer 2. The theoretical distribution of the mass spectrum of the iridium-containing degradation products, P2 and P4, agreed with our experimental data. These results confirmed that the degradation products originated from the emitter molecules. Accordingly, we have revised the manuscript and the Supplementary Information.

3rd paragraph in page 16

(...) For example, the peak intensity corresponding to P3 increases by 5.7% for the LT50 device and 8.8% for the LT30 device. The isotope distributions of the iridium-containing primary products P2 and P4 were further analyzed. The result shows that both degradation products assigned as P2 and P4 ($m/z = 1293.63, 1015.47$) contained iridium atoms, proving that P2 and P4 were the degradation products of the emitter molecule (Fig. S10). This indicates that the electron-induced degradation reactions indeed occurred inside the EML under operational conditions. These degradation reactions chemically bleach the emitter molecule, and the resulting degradation products can play as charge traps and exciton quenchers. [40] Therefore, the degradation reactions proposed here are likely to be the main cause of the irreversible degradation of the device.

1st paragraph in page 17

(...) The second product with a mass of 1050.52 amu can be formed by the combination of P1, P2, and the host molecule. To validate that the six identified products were the result of degradation reactions in the EML, we examined the intensity changes in the mass spectrum with respect to the depth (Fig. S11). The result shows that all the degradation products were found in the EML. This together with the results of the primary products provides

experimental evidence for the feasibility of the proposed electron-induced degradation reactions in a blue OLED device under real operational conditions.

Supplementary Information

Supplementary Section 6. Mass spectroscopic analysis

(a) Theoretical isotope pattern of P2

(c) Theoretical isotope pattern of P4

Supplementary Figure S10. Isotope distribution of primary degradation products which includes iridium

Supplementary Figure S11. Depth profile of the mass spectrum of the aged device whose luminance dropped to 10% of the initial value (Emitter 3) (a) Mass spectrum peak intensity with respect to the ion beam sputter time. (b) Magnified version of (a) for a clearer view of secondary degradation products P1+P3+host-1H

9. Finally, this reviewer carefully analyzed the manuscript, but was unable to find any conceptual difference between the proposed mechanism and the conventional mechanism involving TPA followed by degradation from hot polarons. The mechanism outlined in this manuscript is actually identical to TPA, except that reductive quenching of an exciton is proposed in place of energy transfer to a polaron. The electron-transfer quenching of excitons was also proposed previously (e.g., Nat. Commun. 2018, 9, 1211). The authors are, therefore, highly recommended to underscore their mechanistic novelty.

Reply:

We are not proposing a new mechanism that is different from TPA; instead, we are focused on elucidating the atomistic details of the degradation reactions of blue emitters, which is caused by electrons and their interaction with excitons, among many causes collectively referred to as TPA. Such atomistic details are critical to design new emitter molecules. To further emphasize our distinctive contribution, we have added the importance of understanding the role of electrons and the atomistic degradation mechanism to the revised manuscript.

2nd paragraph in page 2

(...) In addition, various strategies derived from atomic-level understanding have been applied to avoid exciton-induced degradation reactions. [11, 19, 24-29] (...)

1st paragraph in page 3

(...) For instance, polarons can generate exciton quenchers and participate in the bleaching of emitters via a degradation reaction of consisting materials. [38-41] While a high concentration of holes can instigate radical reactions, bond strength analysis indicates that most covalent bonds in OLED materials are weakened in the presence of electrons, whereas holes make them stronger except C-P bonds. [38, 42] This suggests that electrons are the primary factor behind irreversible degradation reactions, resulting in the creation of more exciton quenchers, charge traps, and further luminance drop. (...)

Response to Review 3.

We thank the reviewer for the careful evaluation and valuable comments that have helped us improve the quality of our manuscript. Our point-by-point responses are attached below, and the manuscript has been revised accordingly. All changes in the manuscript were marked in blue.

This study reports the origin of the short lifetime of blue OLEDs, and the authors concluded that electrons in OLEDs are the origin of degradation. Significant efforts have been made on this issue, as the authors described in the Introduction. Studies on TTA, STA, and TPA have been carried out to clarify this problem. This study by Jaewook Kim et al. is one of these studies.

The authors say, "electron-induced degradation reactions play a critical role in the short lifetime," but they do not show experimental data on OLEDs only containing electrons. This is clear in Fig. 2; the three devices contain 1) exciton+hole+electron, 2) exciton only, and 3) exciton+electron. From here, we cannot say that the critical factor of the degradation is electrons, but rather TPA. This is not a new insight.

Reply:

We are not proposing a new mechanism that is different from TPA; instead, we are focused on elucidating the atomistic details of the degradation reactions of blue emitters, which is caused by electrons and their interaction with excitons, among many causes collectively referred to as TPA. Such atomistic details are critical to design new emitter molecules. In addition, Path 1 in Figure 3 illustrates a mechanism in which degradation occurs solely by electrons without the involvement of excitons. To further emphasize our distinctive contribution, we have added the importance of understanding the role of electrons and the atomistic degradation mechanism to the revised manuscript.

2nd paragraph in page 2

(...) In addition, various strategies derived from atomic-level understanding have been applied to avoid exciton-induced degradation reactions. [11, 19, 24-29] (...)

1st paragraph in page 3

(...) For instance, polarons can generate exciton quenchers and participate in the bleaching of emitters via a degradation reaction of consisting materials. [38-41] While a high concentration of holes can instigate radical reactions, bond strength analysis indicates that most covalent bonds in OLED materials are weakened in the presence of electrons, whereas holes make them stronger except C-P bonds. [38, 42] This suggests that electrons are the primary factor behind irreversible degradation reactions, resulting in the creation of more exciton quenchers, charge traps, and further luminance drop. (...)

Moreover, if the authors' conclusion is proper, the four exciton-only devices (this does not contain electrons!) should show a very long device lifetime. However, the lifetimes are limited even for these exciton-only devices (LT95 =2 – 10 hours even @500 cd/m², Fig. 2b). This shows that the degradation's critical factor differs from electrons.

Reply:

While acknowledging the reviewer's concern, we did not claim that the exciton-induced degradation is insignificant. Of course, excitons contribute a lot to the degradation process as can be seen from exciton-only devices. However, this contribution alone cannot explain the lifetime of real devices as shown in Fig. 2b. In the real operating condition, both excitons and electrons can participate in the degradation process. Thus, it is not possible to examine the contribution of excitons alone in the real device and hence estimate how much excitons alone can induce the degradation reactions in real operating conditions. As only considering both electrons and excitons together, the lifetime becomes close to that of the corresponding real device, indicating that electrons (on top of excitons) play an important role in determining the lifetime of the real device. Our work shows diverse degradation pathways involving electrons as depicted in Fig. 4a. Moreover, the phrase, "electron-induced", does not mean that electrons determine the lifetime, but actually implies all degradation processes involving electrons. To clarify these points, we have elaborated on the role of electrons in TPA, as stated in our response to question 1.

As the authors recognize, changing only one factor without changing all the other experimental conditions is difficult. This makes the situation difficult. For example, in excitons, the singlet and triplet ratios in devices 2) and 3) are different from device 1). The spatial distribution (that is, the density of excitons) is also different. According to the preceding studies, the distinction between singlet and triplet is essential but is not discussed here.

Reply:

We acknowledge the reviewer's valuable comment. We also agree that making the exciton density profiles of EODs and WDs is impossible. This concern is also related to the comments from Reviewers 1 and 2. To address this issue, we redesigned the experimental setups to control the impact of both electrons and excitons in EODs as similar to that of WDs as possible, as we revised the manuscript as follows.

1st paragraph in page 7

The lifetimes of WDs were measured in the constant current mode. Although each device has a similar peak emission wavelength, the overall emission spectra are different. For a fair comparison, their lifetimes were measured under a condition where each device emitted the same amount of energy. The initial radiance was set to 2000 mW/sr/m², ensuring that all devices had a brightness level of 300–600 cd/m². For a direct comparison between EODs and WDs, key conditions affecting the degradation of the emitter molecules should be the same for both setups. However, making all individual conditions identical is impossible because the EODs have no hole injection, and more importantly PL and EL have different exciton density profiles inside the EML. Nevertheless, operating conditions can be controlled so that the impact of polarons and excitons on the device lifetime is similar in both setups. To this end, we imposed the following two conditions on the devices. First, we ran the same amount of

current through the EODs with that of the WDs to ensure that electrons have the same effect on the device lifetime. As described above, we minimized the effect of holes in the WDs by using the electron transport type of host molecules. [44] As a result, most electrons and excitons were at the interface of the electron blocking layer and the EML, that has been measured using an exciton probe layer (Supplementary Section 3). [10] Second, we carefully adjusted the intensity of an LED lamp shining on the EODs, so each EOD had a similar lifetime to that of the corresponding WD at the same current. Imposing this condition was not to make the same exciton density profiles of the EODs as the WDs. Polarons (electrons in our setup) and excitons are the primary variables affecting the lifetime. Since we made the effect of electrons similar by applying the same currents to the EODs and WDs, we can expect the overall effect of excitons on the lifetime to be similar if we equalize their lifetimes. However, the exciton-only EODs have no electrons, so they would have different lifetimes under the same excitation condition with that of the exciton+electron EODs. Therefore, we can investigate how excitons with and without electrons affect the lifetime of the EODs and compare it to the WDs. The operating conditions for each device are given in Supplementary Section 4. Fig. 2b shows the lifetimes of all devices, which were measured until the luminance of each device decreased to 95% of the respective initial value (LT95). The lifetime of exciton-only EODs follows a different trend than that of exciton+electron EODs. For devices with emitters 1 and 4, the lifetimes of exciton-only EODs and exciton+electron EODs are similar. However, for devices using emitters 2 and 3, the lifetimes of the EODs differ significantly depending on the presence of electrons. Notably, for the exciton-only EOD, Emitter 3 has a longer lifetime (7.5 h) than Emitter 1 (6.0 h), whereas the WD lifetime of Emitter 3 (2.5 h) is shorter than that of Emitter 1 (5.3 h). This indicates that the effect of excitons only is not sufficient to explain the lifetime under real operational conditions.

Figure 2b

2nd paragraph in Page 3

(...) Our control experiments showed that exciton-only environments could not explain the lifetime trend of blue PhOLEDs under real operational conditions. Instead, electron-induced degradation significantly affects the device's lifetime. (...)

2nd paragraph in page 19

(...) The operational lifetime of blue PhOLEDs was measured under constant current conditions at an initial radiance of 2000 mW/sr/m². (...)

Supplementary Information

Supplementary Section 3. Charge and exciton distribution of whole device

We used an electron transport type host molecule to minimize the influence of holes on the lifetime of the whole device (WD). [S1] In the EML of WDs, emitter molecules are doped in the host film. As the HOMO energy of the emitter molecules is higher than that of host molecules, the doped emitter molecules can improve the hole mobility in the EML. This requires a deeper investigation into the polaron distribution to ensure that the effect of the hole is well suppressed in the devices. Hence, we measured the exciton distribution in the EML by using the methodology described in the reference [S2]. We fabricated probe devices with the same structure as the WDs except an inserted thin red dye layer inside the EML. Here, Ir(dmpq)₂(acac)[S3] was co-doped at 2 vol% at different positions separated by 10 nm in the EMLs, with a doping layer width of 1 nm. The thin layer emits red light by accepting exciton energy from adjacent excited molecules. Comparing the relative intensity of red light from the probe devices, we measured the exciton densities as a function of the position of the probe layer. The relative intensity is calculated using the following equation.

$$\text{Relative Intensity}(x) = \frac{(\text{Intensity of prove device @ 612 nm})(x) - (\text{Intensity of WD @ 612nm})}{\max\{(\text{Intensity of prove device @ 612 nm})(x)\}}$$

As the charge balances of WDs are similar to each other, we fabricated probe devices for Emitter 1 and 3 as examples. Our data shows that excitons are highly concentrated near the EBL/EML interface and are almost absent in the region farther than 20 nm away from the interface. (Figure S8) The result implies that charge recombination occurs predominantly at the EBL/EML interface, and the holes will be concentrated at the same interface.

Supplementary Figure S8. Relative electroluminescence intensities of the red dopants of probe devices.

Supplementary Section 4. Operational condition of each device

Supplementary Table S3. Operational condition of WD

	Emitter 1	Emitter 2	Emitter 3	Emitter 4
Radiance ($\text{mW sr}^{-1} \text{m}^{-2}$)	2000	2000	2000	2000
Luminance (cd m^{-2})	573	561	487	350
Current (mA)	0.064	0.072	0.079	0.137

Supplementary Table S4. Operational condition of EOD

	Emitter 1	Emitter 2	Emitter 3	Emitter 4
UV lamp intensity (mW)	23.2	22.3	15	18.2
Initial radiance (cd m^{-2})	19138	18087	8991	9000
Current for exciton+electron EOD (mA)	0.064	0.072	0.079	0.137

C-P=O and C-N bonds are weak mostly for electrons (see *Adv. Optical Mater.* 2017, 5, 1600901 and *Adv. Optical Mater.* 2020, 8, 2000102). I recommend the use of other materials that are more resistant to electrons.

Reply:

The materials containing C-P=O bonds were only used in the ETL, not in the EML. The C-N bond is commonly found in various blue OLED materials, including emitter and host materials. Recently reported long-lifetime OLED devices also used emitter and host materials containing the C-N bond. [*Nat. Photonics* 2022, 16, 212-228] Additionally, the host and electron transport layer materials has been reported to be robust against polarons and excitons in the solid state. [*Adv. Sci.*, 2017, 4, 1600502] Instead of focusing on specific bonds vulnerable to the electron, we investigated the robustness of different types of bonds against diverse degradation pathways. For example, all emitters used in this study have the C-N bond, but we have shown that with careful ligand design like emitter 1, it is possible to suppress the electron-induced cleavage of the C-N bond.

The AOM 2017 paper also summarizes the reported device lifetimes. Even nine years ago, the lifetime was 213 h @LT80,1000 nit for an Ir-complex with similar CIE and emission wavelength (*Nat. Commun.* 2014, 5, 5008). It has been further updated now. OLEDs with relatively long device lifetimes have been reported. If the authors' new finding is true, the lifetime of these devices can be further improved.

Reply:

The lifetime of a device depends not only on the type of materials used, but also on the structure of the device. In the 2014 paper referred by Reviewer 3, the lifetime was improved by using a graded dopant concentration profile to suppress exciton accumulation in the EML. If we adopted the same engineering in our paper, we could achieve a longer lifetime than what we currently present. However, our work focuses on elucidating the atomistic mechanism of the emitter degradation, because the atomistic detail can provide valuable insight to design new materials with intrinsically longer lifetimes apart from device engineering.

EOD means electron-only device. So, exciton-only EOD and exciton+electron EOD are meaningless.

Reply:

We acknowledge the reviewer's concern. However, to avoid undesirable confusion for readers familiar with "EOD," we used the conventional name EOD instead of introducing a new term. The term "EOD" refers to devices that only allow electron injection, which has been used in various works for control experiments with excitations. [*Chem. Mater.* 2007, 19(8), 2079-2083; *Jpn. J. Appl. Phys.* 2016, 55, 03DD13; *Adv. Electron. Mater.* 2019, 5(5), 1800708]

REVIEWERS' COMMENTS

Reviewer #1 (Remarks to the Author):

In this current revision NCOMMS-22-51354A, the authors have addressed all points of criticism/ for discussion I raised during my original review. The effort put into this revision is very high and very good. All unclear points have been successfully clarified. I am happy to recommend the resulting revised manuscript for publication.

Reviewer #2 (Remarks to the Author):

Kim et al. did an exceptionally good job in revising their original manuscript. This reviewer carefully went through the response provided by the authors, as well as the revised materials. Although the experimental design for comparing lifetime of WDs and EODs still remains controversial, the authors provided detailed experimental conditions that enable readers to judge the validity of the design. This reviewer understands the difficulty of controlling fluences of photons and charge carriers for direct comparison of the operational stability. The original concerns about mass analyses and computational results have also been alleviated by the revisions. The quantum chemical research is also anticipated to provide valuable information to the society of electroluminescence research. The prediction that de-coordination might be feasible in the MLCT transition state due to the trans-effect is highly useful to creating Ir(III) complex dopants robust against degradation. This reviewer would like to recommend the revised manuscript for publication. However, this reviewer would like to recommend the authors to tone down the claim about the novelty of the electron-induced degradation mechanism. It is well-documented that radical anion species are vulnerable to intrinsic degradation of organic molecules and metal complexes for OLEDs (e.g., Chem. Mater. 2016, 28, 5791; Chem. Mater. 2007, 19, 2079; J. Mater. Chem. C 2022, 10, 10139). Therefore, the conclusion delivered by this research is not conceptually novel. In addition, the intrinsic degradation by reductive bond cleavage was also been reported for an Ir(III) complex (J. Fluorine Chem. 2009, 130, 640). Disentangling destructive effects of excitons and polarons was also precedent (Appl. Phys. Lett. 2017, 111, 113301; ACS Appl. Mater. Interfaces 2018, 10, 5693).

Reviewer #3 (Remarks to the Author):

The purpose of this study is to investigate the origin of the current problem of the device lifetime for blue OLEDs. The CIE coordinate of the current target, BT.2020, is (0.131, 0.046) and that of the NTSC

standard is (0.14, 0.08). Of course, these are severe conditions, but at least CIE $y < 0.2$ and < 0.15 are required for blue and deep-blue, respectively. Long lifetime under such conditions is important.

In the four OLEDs in this study, the larger the CIE y coordinate (the shallower the blue), the longer the lifetime. This is a natural result and not different from the previous findings. In particular, the longest-lived OLED in this study has CIE y of 0.33, which is considerably insufficient. It is probably quite greenish. In the extreme, it is like the comparison of the lifetimes between blue and green OLEDs. Does this study provide essential insight into the compatibility of blue emission and long lifetime?

Although I overlooked the chromaticity point in the previous review, the authors would have known that, I think.

In this regards, it is necessary to make comparisons between OLEDs with similar chromaticity coordinates to give correct guidelines.

REVIEWERS' COMMENTS

Reviewer #1 (Remarks to the Author):

In this current revision NCOMMS-22-51354A, the authors have addressed all points of criticism/for discussion I raised during my original review. The effort put into this revision is very high and very good. All unclear points have been successfully clarified. I am happy to recommend the resulting revised manuscript for publication.

Reply: We appreciate the comment of this reviewer.

Reviewer #2 (Remarks to the Author):

Kim et al. did an exceptionally good job in revising their original manuscript. This reviewer carefully went through the response provided by the authors, as well as the revised materials. Although the experimental design for comparing lifetime of WDs and EODs still remains controversial, the authors provided detailed experimental conditions that enable readers to judge the validity of the design. This reviewer understands the difficulty of controlling fluences of photons and charge carriers for direct comparison of the operational stability. The original concerns about mass analyses and computational results have also been alleviated by the revisions. The quantum chemical research is also anticipated to provide valuable information to the society of electroluminescence research. The prediction that de-coordination might be feasible in the MLCT transition state due to the trans-effect is highly useful to creating Ir(III) complex dopants robust against degradation. This reviewer would like to recommend the revised manuscript for publication. However, this reviewer would like to recommend the authors to tone down the claim about the novelty of the electron-induced degradation mechanism. It is well-documented that radical anion species are vulnerable to intrinsic degradation of organic molecules and metal complexes for OLEDs (e.g., Chem. Mater. 2016, 28, 5791; Chem. Mater. 2007, 19, 2079; J. Mater. Chem. C 2022, 10, 10139). Therefore, the conclusion delivered by this research is not conceptually novel. In addition, the intrinsic degradation by reductive bond cleavage was also been reported for an Ir(III) complex (J. Fluorine Chem. 2009, 130, 640). Disentangling destructive effects of excitons and polarons was also precedent (Appl. Phys. Lett. 2017, 111, 113301; ACS Appl. Mater. Interfaces 2018, 10, 5693).

Reply:

Our current work aims to elucidate that electron-induced degradation reactions are the primary cause of intrinsic degradation in blue PhOLED devices under real operating conditions. We have reviewed the references provided by Reviewer 2 and revised the manuscripts as follows.

As the reviewer pointed out, OLED materials are known to be susceptible to electrons. We have included the references provided by the reviewer in the first paragraph on page 3, which introduces the previous research on the vulnerability of OLED materials to electron-induced degradation.

1st paragraph, page 3

(...) For instance, polarons can generate exciton quenchers and participate in the bleaching of emitters via a degradation reaction of consisting materials. [38-41, *Chem. Mater.* 2007, 19, 2079] While a high concentration of holes can instigate radical reactions, bond strength analysis indicates that most covalent bonds in OLED materials are weakened in the presence of electrons, whereas holes make them stronger except C-P bonds. [38, 42, *J. Mater. Chem. C* 2022, 10, 10139] (...)

The paper "*J. Fluorine Chem.* 2009, 130, 640" analyzes the degradation products of Ir(III) complex (Firpic) in aged devices. Although this work revealed several degradation products generated from the Ir(III) complex, the energy source of the relevant reaction of each degradation product was not identified. Other researchers reported that exciton-induced chemical reactions cause the detachment of the picolinate ligand from Firpic. (*Chem. Mater.* 2019, 31, 2269–2276, *Chem. Mater.* 2019, 31, 2277-2285) The reviewer also highlighted two papers (*Appl. Phys. Lett.* 2017, 111, 113301; *ACS Appl. Mater. Interfaces* 2018, 10, 5693) that present a method to experimentally separate two crucial factors that determine the external quantum efficiency of the device: the exciton formation efficiency (η_{EF}) and the photoluminescence efficiency (η_{PL}). These works analyzed the luminance drop that occurs during device operation in the perspective of the light-emitting mechanism. In contrast, our study examines the causes of the luminance drop in the perspective of the durability of consisting materials. Both excitons and electrons can induce degradation reactions of OLED materials, and the degradation products in aged devices can subsequently reduce both η_{EF} and η_{PL} .

Reviewer #3 (Remarks to the Author):

The purpose of this study is to investigate the origin of the current problem of the device lifetime for blue OLEDs. The CIE coordinate of the current target, BT.2020, is (0.131, 0.046) and that of the NTSC standard is (0.14, 0.08). Of course, these are severe conditions, but at least CIE $y < 0.2$ and < 0.15 are required for blue and deep-blue, respectively. Long lifetime under such conditions is important.

In the four OLEDs in this study, the larger the CIE y coordinate (the shallower the blue), the longer the lifetime. This is a natural result and not different from the previous findings. In particular, the longest-lived OLED in this study has CIE y of 0.33, which is considerably insufficient. It is probably quite greenish. In the extreme, it is like the comparison of the lifetimes between blue and green OLEDs. Does this study provide essential insight into the compatibility of blue emission and long lifetime?

Although I overlooked the chromaticity point in the previous review, the authors would have known that, I think.

In this regards, it is necessary to make comparisons between OLEDs with similar chromaticity coordinates to give correct guidelines.

Reply:

As Reviewer 3 pointed out, green OLEDs are known to have longer lifetimes than those of blue OLEDs. This is commonly attributed to the difference in exciton energies that the emitter molecules have to endure. [*Chem. Sci.* 2017, 8, 7844-7850] In the present work, however, all emitters received

almost the same amount of exciton energy. To ensure each emitter received the same amount of exciton energy during the lifetime measurement, we used the same host material for all devices. We also irradiated each device with the same UV lamp for photoluminescence measurements. Furthermore, we chose the emitters to have a similar T_1 state energy to each other. According to quantum chemical calculations, the Gibbs free energy differences between the S_0 and T_1 states are about 2.5 eV for all emitters. The measured emission wavelengths of all emitters further confirm this similarity. The corresponding electroluminescence spectra peaked around 462 nm, except for Emitter 3. (Supplementary Figure S3) Thus, the emitters are subjected to similar exciton energies. It should be noted that the high CIE y coordinate in OLED devices using Emitter 1 and 2 results from the broadening of their emission spectra rather than different exciton energies. For comparison of the T_1 state energy values for each emitter, we have included the Gibbs free energy difference between the S_0 and T_1 states in the Supplementary Information. Consequently, the longer lifetime of the device with the larger CIE y coordinate is not simply due to the lower exciton energies (in fact, all devices had similar exciton energies), which is apparently different from the reason why green OLEDs have longer lifetimes than those of blue ones.

Supplementary Information

Supplementary Table S6. The energy profile of the Ir-ligand bond cleavage reaction. The relative Gibbs free energy is computed as the Gibbs free energy difference between the transition structure and the 6-coordinated emitter (ΔG , unit: kcal/mol).

Degradation process		Emitter 1	Emitter 2	Emitter 3	Emitter 4
Neutral $S_0 \rightarrow$ Neutral T_1	0-0 excitation energy	59.22	58.56	57.74	58.50
Neutral $T_1 \rightarrow$ 3MC	Transition state	10.24	10.65	17.30	- ^a
	5-coordinated complex	6.74	7.16	15.81	20.06
Neutral $T_1 \rightarrow$ Triplet 5Ar	Transition state	33.64	34.09	29.90	24.64
	5-coordinated complex	28.20	28.79	23.42	22.60
Anion $D_0 \rightarrow$ Doublet 5Ar	Transition state	60.16	54.12	56.98	58.83
	5-coordinated complex	55.94	50.89	55.76	53.84
Anion $Q_1 \rightarrow$ Quartet 5Ar	Transition state	32.75	46.41	27.35	30.32
	5-coordinated complex	23.88	22.40	27.28	26.03

^a Barrier energy is too small to detect

Even if the emission spectrum is not deep blue, these emitters can still be utilized as deep blue OLED emitters in Top-Emitting OLEDs [*Adv. Mater.* 2010, 22, 5227-5239]. Top-emitting OLEDs can achieve a sharper emission spectrum than that of the emitter due to the microcavity effect. For instance, introducing an emitter (Firpic) with a PL spectrum CIE y coordinate of 0.34 can yield an OLED device with a CIE y of 0.136 [*Appl. Phys. Lett.* 2007, 90, 211109]. Thus, the emitters we employed in this study are fully viable as blue PhOLED emitters.